# Including ash in UKESM1 model simulations of the Raikoke volcanic eruption reveal improved agreement with observations

Alice F. Wells[1], Andy Jones[2], Martin Osborne[2], Lilly Damany-Pearce[1], Daniel G. Partridge[1] and James M. Haywood[1,2]

[1]Faculty of Environment, Science and Economy, Department of Mathematics and Statistics, University of Exeter, Exeter, EX4 4QE, United Kingdom
[2]Met Office, Exeter, EX1 3PB, United Kingdom

*Correspondence to*: Alice F. Wells (afw207@exeter.ac.uk)

**Abstract.** In June 2019 the Raikoke volcano located in the Kuril Islands, northeast of Japan, erupted explosively and emitted approximately $1.5\text{Tg} \pm 0.2$ Tg of $SO_2$ and $0.4 - 1.8$ Tg of ash into the upper troposphere and lower stratosphere. Volcanic ash is usually neglected in modelling stratospheric climate changes since larger particles have generally been considered to be short-lived in terms of their stratospheric lifetime. However, recent studies have shown that the coagulation of mixed particles with ash and sulfate is necessary to model the evolution of aerosol size distribution more accurately. We perform simulations using a nudged version of the UK Earth System Model (UKESM1) that includes a detailed 2-moment aerosol microphysical scheme for modelling the oxidation of sulfur dioxide ($SO_2$) to sulfate aerosol and the detailed evolution of aerosol microphysics in the stratosphere. We compare the model with a wide range of observational data. The current observational network including satellites and surface based lidars and high-altitude sun-photometers means that smaller-scale eruptions such as Raikoke provide unprecedented detail of the evolution of volcanic plumes and processes, but there are significant differences in the evolution of the plume detected using the various satellite retrievals. These differences stem from fundamental differences in detection methods between e.g. lidar and limb-sounding measurement techniques and the associated differences in detection limits and the geographical areas where robust retrievals are possible. This study highlights that, despite the problems in developing robust and consistent observational constraints, the balance of evidence suggests that including ash in the model emission scheme provides a more accurate simulation of the evolution of the volcanic plume within UKESM1.

## 1 Introduction

Throughout history large explosive volcanic eruptions have resulted in periodic perturbations to the climate. Explosive volcanic eruptions frequently emit a combination of gases, including sulfur dioxide ($SO_2$) and volcanic ash into the UTLS (Upper Troposphere-Lower Stratosphere) where the $SO_2$ oxidises resulting in the formation of secondary sulfate aerosols. Sulfate aerosols in the stratosphere have a residence time of several months to a few years (e.g. Robock, 2000; Langmann, 2014; Jones et al., 2017) due to limited wet and dry deposition rates (Kloss et al., 2021). Sulfate aerosols are primarily reflective and enhance the scattering of shortwave solar radiation, increasing the albedo of the planet and thus exert a cooling effect on

the Earth's climate system (e.g., Robock, 2000; Gordeev, 2014). The extent of their impact upon the climate is dependent on a multitude of parameters, including the magnitude of the emission, location of the volcano, the injection altitude, and the composition of the plume (e.g. Jones et al., 2017).

In June 1991 Mount Pinatubo injected an estimated 10-20 Tg of $SO_2$ into the lower stratosphere (Bluth et al., 1992; Dhomse et al., 2014) causing potentially the largest aerosol perturbation to the stratosphere in the 20[th] century and resulting in average lower tropospheric global temperatures cooling by around 0.5°C across a period of nearly two years (McCormick et al., 1995; Guo et al., 2004). Whilst there has not been another volcanic eruption since Pinatubo to have such a significant impact on the global climate in subsequent years, there have been a series of more moderate eruption. Kasatochi in Alaska erupted in August

2008, injecting an estimated 0.9 – 2.7 Tg of $SO_2$ (Corradini, et al., 2010; Kravitz et al., 2010; Karagulian et al., 2010). The following year in June 2009, Sarychev Peak on the Kuril Islands was estimated to have injected $1.2 \pm 0.2$ Tg (Haywood et al., 2010) and in June 2011 Nabro in Eritrea erupted injecting around 1.3 – 1.5 Tg of $SO_2$ (Clarisse et al., 2012). These eruption estimates for the three volcanic eruptions were later refined using a consistent algorithm; 0.9 Tg, 1.5 Tg and 0.5 Tg loadings were derived above 10km using the Michelson Interferometer for Passive Atmospheric Sounding (MIPAS) on Envisat

(Höpfner et al., 2015). Haywood et al., (2014) estimate that over the period 2008 – 2012 these smaller volcanic eruptions contributed to between -0.02 and -0.03 K of cooling at the Earth's surface.

While these volcanic eruptions injected an order of magnitude less $SO_2$ into the stratosphere than Pinatubo, monitoring the transport, evolution and dispersion of volcanic plumes allows an assessment of the performance of global climate models in

representing stratospheric sulfate plumes, and allows improvements to be made in key processes. The much-improved observational network compared to that which observed the Pinatubo eruption, which includes satellite observations of both $SO_2$ (e.g. Cai et al., 2022) and sulfate aerosol (Lee et al., 2009), surface based lidars (e.g. Chouza et al., 2020), high-altitude sun-photometers (e.g. Toledano et al., 2018), and periodic balloon-borne observations (e.g. Jégou et al., 2013) means that observations of these smaller-scale eruptions provide unprecedented detail of the evolution of volcanic plumes and processes.

The validation and improvement of representation of volcanic plumes within global climate models leads to a better understanding of their associated cooling impacts. Such synergy between observations and models also provides a means to assess the uncertainties associated with proposed stratospheric aerosol injection climate intervention strategies that have recently been suggested as a method to ameliorate the worst impacts of climate change (e.g. Lawrence et al., 2018).

This study examines the impact of the 2019 eruption of Raikoke. Almost exactly a decade after the eruption of Sarychev Peak (12[th] June 2009, 48.1°N, 153.2°E) on 21[st] June 2019 at 1800 UTC, a neighbouring volcano – Raikoke (48.3°N, 153.2°E) – started to erupt, generating a series of distinct explosive events, and emitting a plume of ash and $SO_2$ into the stratosphere. During this period, it is estimated that it injected around $1.5 \pm 0.2$ Tg of $SO_2$ (Muser et al., 2020; Kloss et al., 2021; De Leeuw et al., 2021) and 0.4 – 1.8 Tg of ash (Bruckert et al., 2022) into the stratosphere signifying the largest volcanic emission of $SO_2$

since the Nabro eruption in 2011. The resultant volcanic aerosol plume was detected at altitudes ranging between 11 to 20km by the TROPOMI instrument (Hedelt et al., 2019; Vaughan et al., 2021) and at similar altitudes by other satellite instruments (e.g., Gorkavyi et al., 2021, Kloss et al., 2021), although altitudes as high as 26km have been inferred in isolated lidar measurements (e.g. Chouza et al., 2020). These findings indicate that a significant portion of the volcanic plume was injected into the stratosphere. Gorkavyi et al. (2021) found that the peak sulfate aerosol extinction occurred around 1.5 months after

the eruption date with an $SO_2$ e-folding lifetime of approximately 19 days. Previous studies looking at similar volcanic eruptions have found $SO_2$ *e*-folding times at a similar scale. For example, once the retrieval minimum detection threshold had been accounted for, Haywood et al., (2010) determined an e-folding for $SO_2$ from the Infrared Atmospheric Sounding Interferometer (IASI) for the Sarychev Peak eruption of around 20-22 days.

Less than a week after the eruption of Raikoke a second volcanic eruption occurred – Ulawun (5.1°S, 151.3°E) – on 26[th] June 2019 and again on 3[rd] August 2019. It is estimated to have injected around 0.14 Tg $SO_2$ into the stratosphere during the first explosive eruptive phase and a further 0.2 Tg $SO_2$ during the second phase of the eruption (Kloss et al., 2021).

The Raikoke and Ulawun eruptions were both well observed by a series of satellite instruments and ground-based measurement
stations. Satellite observations include the Ozone Mapping Profiler Suite (OMPS) Nadir Mapper (NM) (Yang, 2017) and Limb Profiler (LP) (Taha, 2020) and the Cloud-Aerosol Lidar with Orthogonal Polarization (CALIOP) (Winker et al., 2009) while surface observations include those from high altitude AErosol RObotic NETwork sites (AERONET; Holben et al., 1998). Although the perturbations to the Earth's radiation budget and near-surface temperature from moderate volcanic eruptions, such as Raikoke, are unlikely to be detectable owing to the small signal-noise ratio, these impacts can be estimated from Earth
System models. The Raikoke eruption was the largest volcanic stratospheric injection of $SO_2$ since the OMPS satellite was launched in late 2011 providing an excellent opportunity to assess the skill and the limitations of the UK Earth System Model (UKESM1; Sellar et al., 2019) in simulating the evolution of the atmospheric distributions of $SO_2$ and sulfate aerosol.

Recent studies have drawn attention to the influence of ash on self-lofting and the evolution of the volcanic plume (e.g., Muser
et al., 2020; Kloss et al., 2021). Volcanic ash is usually neglected in the modelling the impact of eruptions on the stratosphere and climate since larger particles (radii r > 1µm) would be short lived owing to their considerable fall-speed (Niemeier et al., 2021, 2009; Stenchikov et al., 2021). However, Zhu et al., 2020 showed that, to produce the evolution of the size distribution following the Kelud eruption in 2014, the coagulation of internally mixed ash and sulfate particles is necessary. They also found that after this eruption super-micron sized ash particles with an estimated density (0.5 g cm$^{-3}$) corresponding to pumice
were the main component of the volcanic aerosol layer. This is in contrast to the assumed density (~ 2.3 g cm$^{-3}$) of ash within current models. Including ash emissions in model simulations has been found to alter the dynamics of sulfate aerosol formation (Shallcross et al., 2021; Stenchikov et al., 2021) including prolonging the lifetime of stratospheric aerosol optical depth (sAOD) (Kloss et al., 2021). Stenchikov et al. (2021) and Abdelkader et al. (2023) agreed that when modelling the Pinatubo eruption,

including volcanic ash increases the radiative heating during the first week after the eruption and results in the lofting of the
aerosol.

Muser et al. (2020) examined the impacts of aerosol-radiation interactions and aerosol dynamics on volcanic aerosol dispersion. They showed that during the first days after the Raikoke eruption, the absorption of solar radiation caused by the presence of ash had a significant impact on the aerosol dispersion, producing a self-lofting effect on the plume. Over the course
of 4 days after the eruption, the maximum cloud top height rose more than 6km (Muser et al., 2020). Within a few weeks the volcanic plume dispersed across the Northern Hemisphere (NH) and was continually observed months after the eruption. The radiative self-lofting could explain some of the differences between observations and model simulations which did not account for ash in previous studies (Haywood et al., 2010; Kloss et al., 2021) since the self-lofting effect would result in a greater fraction of the plume in the stratosphere and subsequently result in a longer residence time. Stenchikov et al. (2021) also found
in model experiments of the Pinatubo eruption that during the first week after the eruption $SO_2$ and sulfate plumes in the presence of ash rose 7km above injection. It has been shown that whilst the ash does not provide a direct climate impact and the aerosol optical depth decreases quickly, the impact of the ash on the dynamical lofting of the plume is very important for the mass of the aerosol remaining in the stratosphere (Stenchikov et al., 2021).

Several studies have also discussed the influence of self-lofting caused by the presence of soot from intense forest fires (e.g. Fromm et al., 2005; Peterson et al., 2018; Christian et al., 2019; Damany-Pearce, 2022). Of particular note are the studies of Ansmann et al., (2021) and Ohneiser et al., (2021) who used state-of-the-art lidar retrievals mounted on an ice-breaker ship that drifted in the Arctic circle during winter 2020 to infer that biomass burning smoke from intense Siberian wildfires was present in significant quantities in the lower stratosphere, although these results remain contentious (e.g. Boone et al., 2022).
We restrict our study to simulations of the Raikoke eruption including and excluding volcanic ash.

The aim of this study is to assess the modelled volcanic emissions that best represent the observed Raikoke volcanic eruption. We compare observations with a model simulation injecting only $SO_2$ with a model simulation with both $SO_2$ and ash. We use these to establish both how well UKESM1 performs in modelling the volcanic plume and to determine if the inclusion of ash
in modelled volcanic emissions leads to a better agreement with observations. In Sect. 2 we introduce the observational data sets used and the differences in retrieval techniques. Furthermore, we provide a description of UKESM1, and the simulation set up in Sect. 3. In Sect. 4 we present the results and discussion before conclusions are drawn in Sect. 5.

## 2. Observational data and quality assurance

### 2.1 CALIPSO

The Cloud-Aerosol Lidar and Infrared Pathfinder Satellite Observation (CALIPSO) satellite (Winker et al., 2009) combines an active lidar instrument with passive infrared and visible imagers to analyse the vertical structure and properties of thin cloud and aerosols. The Cloud-Aerosol Lidar with Orthogonal Polarization (CALIOP) instrument is a dual-wavelength (532nm and 1064nm) polarization-sensitive lidar which provides high-resolution vertical profiles of aerosols and clouds. The aerosol profile products are reported at a uniform spatial resolution of 5km horizontally. The vertical resolution of the data varies as a

function of altitude, with 60m vertical resolution in the troposphere and 180m vertical resolution in the stratosphere.

This study uses quality-assured (QA) daily averaged vertical profiles of aerosol extinction (km$^{-1}$) at 532nm from the Version 4.20 CALIOP 5km Level 2 Cloud and Aerosol Profile data product. Aerosol extinction coefficients are reported for each bin in which aerosol particulates were detected, those in which no aerosols were detected contain fill values (-9999) and those in

which the extinction retrieval failed were assigned a fill value of -333. These were mapped to a 1° x 1° latitude/longitude spatial grid whilst maintaining the original vertical profile. Quality control procedures were applied to the data in a similar fashion to those implemented in Campbell et al. (2012) which includes quality assurance on the stability of the retrievals and accounts for missing data when retrieval stability fails. The stratospheric aerosol optical depth is calculated by integrating over altitudes above the observed tropopause.


Active lidar retrievals, such as those obtained by CALIOP, are susceptible to solar background contamination which results in poorer performance in day-time conditions resulting in different minimum detection thresholds. It is estimated that the night-time threshold is 0.012 km$^{-1}$ and the day-time thresholds is 0.067 km$^{-1}$ (Toth et. al., 2018). The day-time detection threshold results in a column integrated underestimate of the AOD, and it has been found that it is unable to detect around 50% of aerosol

profiles when the AOD is less than 0.1 (Toth et. al., 2018). For this reason, this study only uses the night-time retrievals to create the daily average extinction values to avoid an underestimated daily average. However, utilising only the night-time profiles leads to large areas of missing data, specifically at high latitudes during the Northern Hemisphere summer where areas experience 24 hours of sunlight. At most northern latitudes (60°N – 90°N) the CALIOP night-time profiles miss the initial peak in aerosol between 30 and 100 days after the eruption. Whilst the maximum night-time sAOD is approximately 65%

greater than the peak daytime sAOD, evaluating only the night-time profiles could influence the timing of the sAOD peak. This is discussed further in Sect. 4.

### 2.2 OMPS

The Suomi National Polar-orbiting Partnership (NPP) is a weather satellite which was launched in 2011 with five imaging systems, including the Ozone Mapping and Profiler Suite (OMPS), a series of instruments comprised of back-scattered

ultraviolet radiation sensors. These sensors measure and monitor atmospheric trace gases, aerosols, surface reflectance and cloud-top pressure. There is global spatial coverage providing a good opportunity to evaluate the plume at high latitudes. Retrieved profiles have a vertical resolution of approximately 1.8km, with profiles being measured from the ground to about 80km (Taha et. al., 2021).

**2.2.1: OMPS-NM:** The OMPS Nadir Mapper (NM) measures backscattered UV radiance spectra between 300-380nm and whilst it is primarily designed to measure global total ozone, the $SO_2$ vertical column amount can be derived from the hyperspectral measurements of the OMPS-NM instrument. This study utilises the $SO_2$ Level 2 orbital products to assess the distribution of $SO_2$ after the eruption. A QA scheme is applied to daily profiles of total column $SO_2$ data, retrieved with a prescribed lower stratospheric profile centred at 16km above the surface. The screening includes discarding pixels when the solar zenith angle is greater than approximately 88° or viewing zenith angle is greater than approximately 70° (Yang, 2017).

**2.2.2: OMPS-LP:** In addition to the CALIOP aerosol extinction data, we utilise retrievals of the vertical aerosol extinction coefficient ($km^{-1}$) from the OMPS Limb Profiler (LP). The OMPS-LP is a passive sensor which looks back along the orbit track at the Earth's limb and records atmospheric spectra which are used to retrieve aerosol extinction coefficient profiles from the lower stratosphere (10 – 15km) to the upper stratosphere (55km). Aerosol extinction measurements are provided at wavelengths ranging between 510 – 997nm at 1km altitude intervals between the surface and 40.5km. This study utilises the V2.0 data measured at 869nm, which have been found to be the best OMPS-retrieved wavelength relative to SAGE III (Taha et al., 2021). Relative to V1.5 data, an improved cloud screening criterion is used in V2.0, which does not remove fresh volcanic plumes and allows us to use the filtered Retrieved Extinction Coefficient data product which removes the influence of polar stratospheric clouds (Taha et al., 2021). As with the CALIOP data, quality control procedures are applied to the OMPS-LP data. These include removing values where the cumulative residual error exceeds a threshold value, when the single scattering viewing angle exceeds 145° and where the derived aerosol scattering index is less than 0.01 (Johnson et al., 2020).

Due to the viewing geometry and sensitivity of the instrument, OMPS-LP can detect aerosol extinction coefficient values down to a minimum value of $1 \times 10^{-5}$ $km^{-1}$ (Johnson et al., 2020), which is far more sensitive than CALIOP (Section 2.1.1). However, OMPS-LP experiences loss of sensitivity of short wavelength radiances to aerosols, caused by Rayleigh scattering and aerosol attenuation of the limb scattered radiation, which is most pronounced below ~17km and especially in the southern hemisphere (Johnson et al., 2020). The retrieval issues described here, particularly the altitude sensitivity, have a significant impact on our study since it was estimated that the initial plume reached altitudes of between 11 and 20km (Vaughan et al., 2021; Osborne et al., 2022), therefore we utilise the 869nm wavelength data and scale it to 532nm to compare to CALIOP. However, once the self-lofting of the plume occurs and it is dispersed over the northern hemisphere it is expected that the increased sAOD becomes more readily detectable by OMPS-LP (Hirsch and Koren, 2021), while it becomes less detectable or undetectable by CALIOP due to CALIOP's significantly higher minimum detection threshold. This is examined in more detail in Sect. 4.

## 2.3 AERONET

AERONET provides whole atmosphere AOD observations at a series of sites distributed across the globe providing a good global coverage of ground-based remote sensing data. One of these sites, the Mauna Loa Observatory (MLO), located at 3397m above sea level in Hawaii, provides an excellent opportunity to monitor stratospheric events. The measurement site is generally removed from the influence of pollution sources and is located at an altitude higher than most tropospheric aerosols. This provides an opportunity to retrieve ground-based observations of the stratosphere using sun-photometry with minimal

tropospheric influences. MLO has been monitoring the stratospheric aerosol layer with lidars since 1975 (Barnes and Hofmann, 1997) providing a long-term historical record and previous studies have demonstrated that aerosol from the Raikoke plume was readily detectable (Chouza et al., 2020). Rather than lidars, this study uses daily Level 2 AOD AERONET retrievals measured at 500nm which are automatically cloud-cleared and quality assured.

## 3. Model simulations

### 3.1 UKESM1

UKESM1 is the latest UK Earth system model, described by Sellar et al. (2019). UKESM1 consists of the HadGEM3 coupled physical climate model with additional interactive components including modelling key biogeochemical processes (Yool et al., 2013), tropospheric and stratospheric chemistry (Archibald et al., 2020), aerosols (Mann et al., 2010) and sea-ice (Ridley et al., 2018). The atmosphere has a horizontal resolution of 1.25° latitude by 1.875° longitude with 85 vertical levels and a

model top at around 85km (Storkey et al., 2018). The StratTrop chemical mechanism used in UKESM1 is described by Archibald et al., (2019). This merged stratospheric and tropospheric scheme simulates interactive chemistry from the surface to the top of the model, including oxidation reactions responsible for sulfate aerosol production (Sellar et al., 2019).

Atmospheric composition in UKESM1 is simulated by the UK Chemistry and Aerosols (UKCA) sub-model. One of the main

components of UKCA is the GLOMAP-mode modal aerosol scheme described in Mann et al., (2010) and Mann et al., (2012). GLOMAP-mode is a two-moment aerosol microphysics scheme which simulates speciated aerosol mass and number across five lognormal size modes, 4 soluble modes (nucleation, Aitken, accumulation, and coarse modes) and one insoluble Aitken mode. The prognostic aerosol species represented are sulfate, black carbon, organic carbon, and sea salt, with species within each mode treated as an internal mixture. The size ranges of covered by each mode are shown in Table 1.


| Aerosol mode | Radii (nm) | $\sigma_g$ |
|---|---|---|
| Nucleation sol. | $0 - 5$ | 1.59 |
| Aitken sol. | $5 - 50$ | 1.59 |

| | | |
|---|---|---|
| Accumulation sol. | 50 – 250 | 1.40 |
| Coarse sol. | 250 – 5000 | 2.00 |
| Aitken insol. | 5 – 50 | 1.59 |

**Table 1:** The aerosol size distribution in GLOMAP-mode including the aerosol modes represented, the range of radii that these include and their geometric standard deviation.

This configuration for UKESM1 and UKCA has been used in many studies to model the evolution of sulfur dioxide into sulfate aerosol, most recently by Visioni et al. (2023) and Bednarz et al (2023), who examined the evolution and climate impacts of the sulfur dioxide and sulfate plume under continuous stratospheric injection three different models. In terms of the distribution and sulfate aerosol optical depth, the resultant plume from UKESM1 when injecting at the most northerly latitude in that study (30°N) was broadly consistent with that from both CESM2 and GISS models, lending confidence to the ability of UKESM1
to accurately model mid-high latitude stratospheric injections.

In addition to the UKCA aerosol components, mineral dust is included as an externally mixed aerosol via the CLASSIC (Coupled Large-scale Aerosol Simulator for Studies In Climate) six-bin scheme detailed in Woodward (2011), which represents mineral dust with diameters ranging from approximately 0.06 to 60µm. This scheme is modified to provide a suitable
proxy for volcanic ash as detailed in section 3.2.

### 3.2 Simulations and reference wavelengths for model/observation intercomparisons

Simulations of the Raikoke and Ulawun eruptions were performed by nudging horizontal winds towards ERA5 reanalysis data to produce relevant meteorological conditions for the respective period using the atmosphere-only configuration of UKESM1.
Nudged simulations were performed without any explosive volcanic emissions as a control (CNTL), $SO_2$ emissions only (SO2only) and with $SO_2$ and ash emissions (SO2+ash). The Raikoke eruption was initiated for the 24-hour period starting at 00:00 UTC on 21[st] June 2019, with a constant emission rate. Emissions were injected into a single column within the model framework at two injection altitudes, a lower "tropospheric" injection at 10km and an upper "stratospheric" injection at 13-15km where the emissions were distributed equally across the altitude range. A total of 1.5 Tg $SO_2$ (Kloss et al., 2021) was
injected and, for the SO2+ash scenario, 1.1 Tg of ash (Muser et al., 2020) was also injected at the same altitudes as $SO_2$. Injection altitudes and masses of $SO_2$ and ash are consistent with observations and those found in the literature (Muser et al., 2020; Kloss et al., 2021; De Leeuw et al., 2021). The emission profile was weighted so that 80% was emitted into the stratosphere and the remaining 20% into the troposphere, based on observations of the $SO_2$ vertical profile (De Leeuw et al., 2021; Osborne et al., 2022).


Emissions of ash are implemented by adapting the Woodward (2011) bin scheme for mineral dust as a suitable proxy. The justification for doing this stems from the fact that the refractive indices and size distributions are similar (e.g. Millington et al., 2012; Johnson et al., 2012; Osborne et al., 2022), although it is recognised that substantial inter-eruption and inter-eruption-phase variability in volcanic ash refractive indices occurs (e.g. Millington et al., 2012; Turnbull et al., 2012). The ash is

moderately absorbing with a refractive index of $1.52 + 0.0015i$, based on the mineral dust from Balkanski et al., (2007) with the medium level of hematite (1.5%). Volcanic ash size distributions were based on observations of the Eyjafjallajökull eruption presented in Johnson et al. (2012) fitted by lognormal distributions (Table 5 of Johnson et al., 2012). The lognormal parameters for the overall mean aerosol size distribution include a volume geometric mean diameter of 3.8 and standard deviation of 1.85. Transport and deposition of dust is as described in detail in Woodward (2001) with improvements to the

emission scheme and refractive index data described in Woodward (2022). In the current configuration mineral dust aerosol is simulated independently of other aerosol species using the CLASSIC dust scheme (Bellouin et al., 2011). Mineral dust can therefore be considered externally mixed with the GLOMAP aerosols.

For both SO2only and SO2+ash simulations, the Ulawun eruptions are simulated by UKESM1 with an $SO_2$ only injection (no

ash emissions) and are initiated on 26[th] June and 3[rd] August 2019. Injection altitudes for Ulawun were 13-17km (26[th] June) and 14-17km (3[rd] August) using 0.14 Tg and 0.30 Tg $SO_2$ respectively (Kloss et al., 2021).

To facilitate the intercomparison of the observations and the model simulations all datasets were scaled to 532nm. Mie scattering calculations were performed using the Mie scattering code within SOCRATES (Suite Of Community RAdiative

Transfer codes based on Edwards and Slingo; Edwards and Slingo, 1996) to generate the single scattering properties of volcanic aerosol at a range of specified wavelengths. Note that both the Mie scattering code and SOCRATES are used in the radiative transfer code within UKESM1. The size distribution of the volcanic aerosol used here was based upon the bimodal lognormal size distribution for a moderate loading volcanic eruption (SPARC, 2006). Specific extinction coefficients were calculated to allow all observational and model data to be scaled to one consistent wavelength: 532nm.


## 4. Results

In analysing the results, we examine the injection of the $SO_2$ and ash through to the gas phase oxidation of the $SO_2$ to sulfate aerosol and the ultimate deposition of the aerosol until the stratospheric perturbation is no longer detectable. Different aspects are investigated including the geographic distribution and temporal evolution of the $SO_2$ and sulfate aerosol and the latitudinal

distribution and vertical profile of the sulfate aerosol.

## 4.1 Observed and modelled SO₂ including and excluding volcanic ash

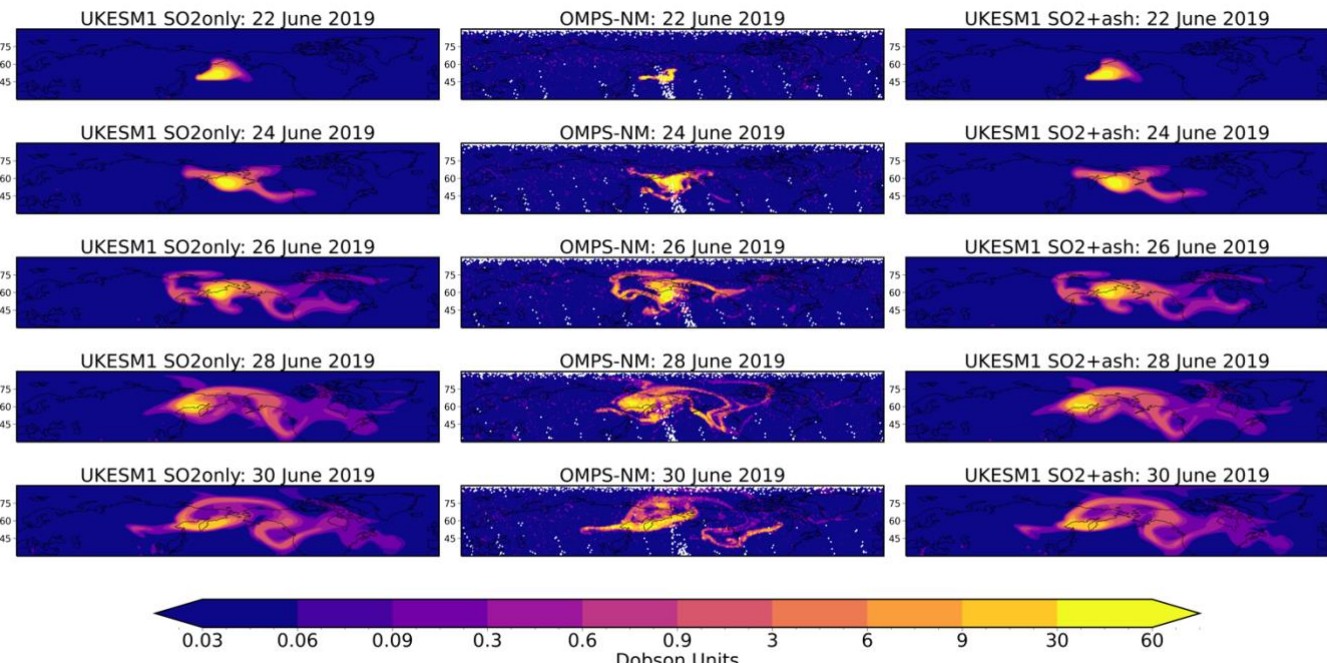

**Figure 1:** Geographic evolution of column integrated SO$_2$ plume in Dobson Units (DU) derived from OMPS-NM lower stratospheric profile (centre) UKESM1 SO2only (left) and SO2+ash (right) for the period 22–30th June 2019. We remove the long-term background SO$_2$ burden derived from OMPS-NM for the years 2013–2018 from those for 2019 to provide a stratospheric perturbation for the observations. Similarly, we remove the impacts of background stratospheric aerosol from the model simulations by subtracting the stratospheric sulfate burdens from the CNTL simulation from those for SO2only and SO2+ash. The OMPS-LP background and CNTL SO$_2$ burden are shown in Fig. S1.

Figure 1 shows the evolution of the SO$_2$ cloud from 22nd June through to the 30th June 2019. The middle column shows the OMPS-NM observations, with the UKESM1 SO2only and SO2+ash simulations on the left and the right respectively. We see that the position and timing of the plume is relatively well modelled throughout this period in both simulations. There is little difference in the spatial pattern of the SO$_2$ plume in both simulations, making it difficult to determine based upon SO$_2$ alone which simulation best represents the observations. Qualitatively, the spatial pattern of the plume is better represented in both the model simulations from 26th June onwards, with both the easterly and westerly parts of the plume well modelled. The largest difference between the observations and the model simulations is seen on 22nd June where the model is initially much more diffuse. This is due to the eruption taking place at 18:00 UTC 21st June and it was inherently explosive and sporadic in nature compared to the smooth injection rates that are assumed in the model. This could explain why the modelled plume does not represent the observations as well during the first two days after the eruption. Similar conclusions have been found in a

recent modelling study that use the CALIOP lidar to assess the fidelity of an operational dispersion model in determining the evolution of a large pyro-cumulonimbus event (Osborne et al., 2022; their Figure 7). Since the model does a reasonable job at representing the shape and distribution of the plume after a few days, and our objectives are to assess the general model performance over a period of many months, we retain our simplified emission period and assumed vertical profile. Higher resolution modelling assessments using the Met Office Numerical Atmospheric-dispersion Modelling Environment (NAME),

that are more appropriate for operational monitoring of volcanic plumes for the first few weeks after the eruption for the purposes of aviation safety are available in de Leeuw et al., (2021) and Osborne et al., (2022).

In the first few days after the eruption the $SO_2$ plume becomes trapped within a cyclonic circulation across Eastern Russia and Alaska (e.g. Osborne et al., 2022). We can observe this feature in both the observations and the model simulations. However,

as observed in other similar studies of other volcanic eruptions (e.g., Haywood et al., 2010) the model $SO_2$ plume becomes more diffuse than observations over time. We can see that as the plume evolves, despite the model capturing the general position, the model overestimates the tail crossing North America and underestimates the magnitude of the plume over Russia. This can also be due to the instrumental detection limits, where the plume has become so diffuse it becomes undetectable. We also notice that the model is generally more diffuse than the observations, which has been observed with other numerical

transport schemes (e.g. de Leeuw et al., 2021), which could contribute to the differences seen in the modelled and observed tails.

To provide a more quantitative analysis of the geographic evolution we employ a dichotomous forecast style analysis. A contingency table is a simple way to identify the frequency of "yes, an event will happen" and "no, the event will not happen"

forecasts and occurrences. For this analysis we treat the model simulation as the forecast and the observations as the occurrence for each grid box on each day. There are four combinations of simulations and observations, "*hits*" – the model simulates the observations correctly, "*model > observations*" – the model overestimates the observations, "*observations > model*" – the model underestimates the observations and "*correct negative*" – both the observations and the model are below a given threshold, 0.3 Dobson Units (DU). In developing the contingency table we consider estimates of error such as timing errors in

synoptic meteorological features that frequently occur in weather forecasting and the fact that the model and observations are not perfectly collocated in time. We therefore assume that the observations are uncertain by a factor of two and use these as the upper and lower bounds. However, we recognise that much more detailed and comprehensive approaches to forecast verification have been developed (e.g. Casati et al., 2008).

Figure 2 presents this analysis for the first 10 days after the eruption. Both simulations show a similar distribution, so we focus on SO2only here. It is clear that "*model > observations*" dominates, with a large tail over North America, as seen in Figure 1. However there are some regions where the model is underestimating the observations which may not have been identified by eye in Figure 1. Over the 20 days after the eruption, approximately the time for the oxidation of $SO_2$ to sulfate aerosol, both

model simulations overestimate the observed plume 52% of the time. We also note that between 15 – 17% of the plume is

correctly modelled within the bounds of the observations for both simulations, with SO2+ash underestimating the observations

3% more than SO2only (Table 2).

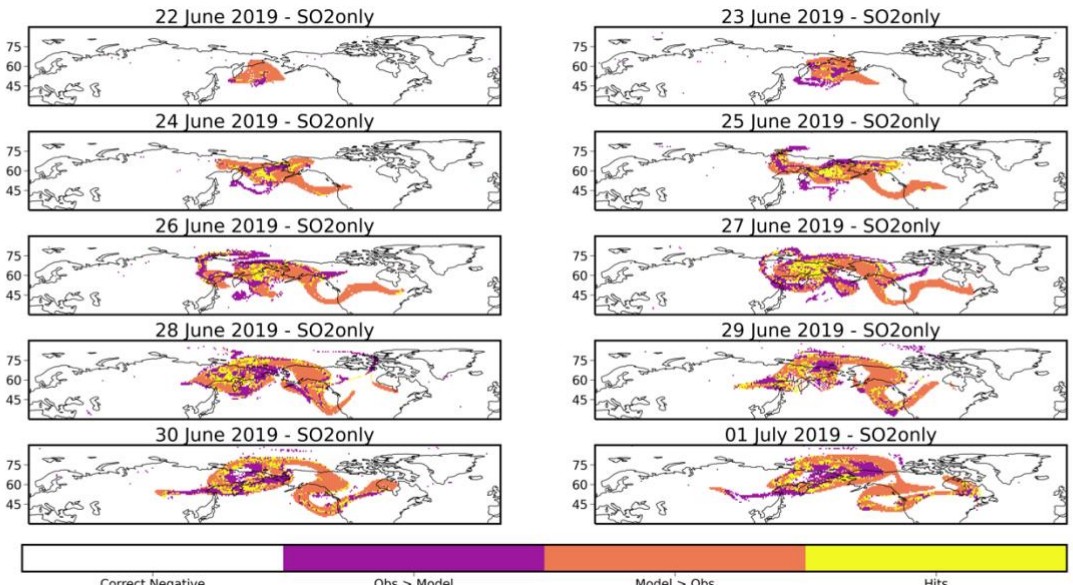

**Figure 2:** Contingency analysis of $SO_2$ plume between OMPS-NM lower stratospheric profile and UKESM1 SO2only for the period 22nd June–1st July 2019. "Correct Negative" occurs at the point where both the model and observation are below 0.3

DU. "Hits" occur at the point where the modelled column $SO_2$ burden is within a factor of two of the observations at that point to allow for timing errors. "Obs > Model" and "Model > Obs" occur when the modelled column $SO_2$ burden is above or below the factor of two limit.

|  | "Hits" | "Observations > Model" | "Model > Observations" |
|---|---|---|---|
| SO2only | 17% | 31% | 52% |
| SO2+ash | 15% | 34% | 51% |

**Table 2:** Contingency analysis of $SO_2$ plume between OMPS-NM lower stratospheric profile and UKESM1 SO2only and

SO2+ash for the 20 days following the eruption on 21st June 2019. "Hits" occur at the point where the modelled column $SO_2$ burden is within a factor of two of the observations at that point to allow for timing errors. "Obs > Model" and "Model > Obs" occur when the modelled column $SO_2$ burden is above or below the factor of two limit.

In Figure 1 we can see that the $SO_2$ plume travels longitudinally and moves towards more northern latitudes, as we would expect from the stratospheric Brewer-Dobson circulation (e.g. Haynes, 2005), and as evidenced from the previous eruption of Sarychev Peak (Haywood et al., 2009; Jégou et al., 2013). Due to this poleward transport, it is unlikely that the Raikoke $SO_2$ plume would travel south of 30°N, particularly in the first few months after the eruption. Hence, to avoid any influence from the Ulawun eruption we take the area weighted average from 30 – 90°N (discussed further in Sect. 4.3) to determine the temporal evolution of $SO_2$ and calculate an $e$-folding time. Figure 3 shows the daily column burden of observed and modelled $SO_2$ after the eruption. The spike seen at approximately day 60 is an artifact from the long-term background $SO_2$ burden. The observations show a peak column burden of 0.76 DU and have an $e$-folding time of 20 days. Model simulations had similar $e$-folding times of 19 and 21 days for SO2only and SO2+ash respectively. This suggests that the oxidation processes are well represented in the UKESM1 model and are very similar to those determined for the Sarychev Peak eruption for the forerunning HadGEM-2 climate model (Haywood et al., 2010). However, in both SO2only and SO2+ash model simulations the peak $SO_2$ column burden is only 0.44 DU, considerably less than that observed by OMPS-NM. The notable difference between the observations and the model is unexpected given that the magnitude of $SO_2$ injected was based on observations (Muser et al., 2020; Kloss et al., 2021; De Leeuw et al., 2021). However, if the amount of $SO_2$ injected into the model simulations were to be increased it would lead to a significant overestimate of sulfate aerosol and sAOD (see later sections). For this reason we do not change the amount of $SO_2$ injected. We do note that the 1.5 Tg of $SO_2$ that we chose to inject into UKESM1 is based upon measurements from TROPOMI and HIMAWARI data and therefore may not exactly correlate with the OMPS-NM data.

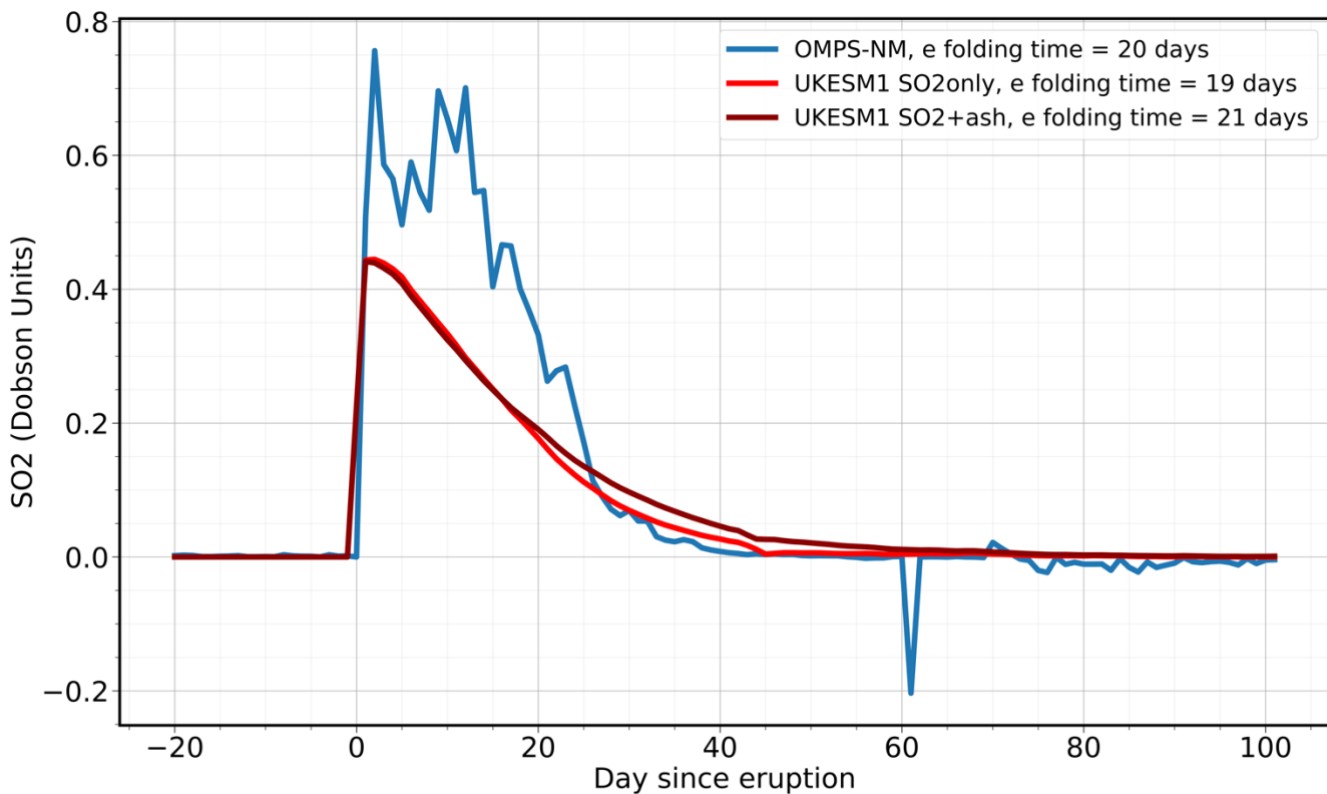

**Figure 3:** Daily perturbation of $SO_2$ in Dobson Units (DU) derived from OMPS-NM lower stratospheric profile (blue), UKESM1 SO2only (red) SO2+ash (dark red). Data averaged across latitudes 30–90° N, weighted by the cosine of the corresponding latitude to ensure data is area weighted. We remove the long-term background $SO_2$ burden derived from OMPS-NM for the years 2013–2018 from those for 2019 to provide a stratospheric perturbation for the observations. Similarly, we remove the impacts of background stratospheric aerosol from the model simulations by subtracting the stratospheric sulfate burdens from the CNTL simulation from those for SO2only and SO2+ash.

### 4.2 Distribution of sulfate aerosol

To investigate the distribution and evolution of the sulfate plume we utilise the CALIOP and OMPS-LP retrieved aerosol extinction integrated above the tropopause to find the perturbed sAOD. We firstly investigate the temporal evolution of the zonal mean sAOD by performing similar analysis to previous studies (e.g., Kravitz et al., 2010; Haywood et al., 2010; Kloss et al., 2021). We compare the evolution of the OMPS-LP and CALIOP retrievals against the UKESM1 SO2only and SO2+ash scenarios, shown in Figure 4. Due to the differences in satellite retrievals discussed in Section 2, there are seasonal gaps in the data from as far south as ~55°N in CALIOP night-time retrievals, due to polar summer and ~65°N in OMPS-LP due to the

lack of daylight hours in NH winter. Additionally, the observations have different minimum retrieval limits (0.012 km$^{-1}$ for CALIOP, 1 x 10$^{-5}$ km$^{-1}$ for OMPS-LP) so to ensure better comparisons we have applied both requirements to both model simulations and scaled all data to a wavelength of 532nm. The CALIOP retrieval only reports the aerosol extinction coefficient for layers in which aerosol particulates were detected above the minimum retrieval limit and uses fill values for the rest of the

profile. Therefore, it is important to note that the CALIOP sAOD could be biased towards large values of aerosol extinction during the first few months after the eruption and towards smaller values after the plume has become more diffuse.

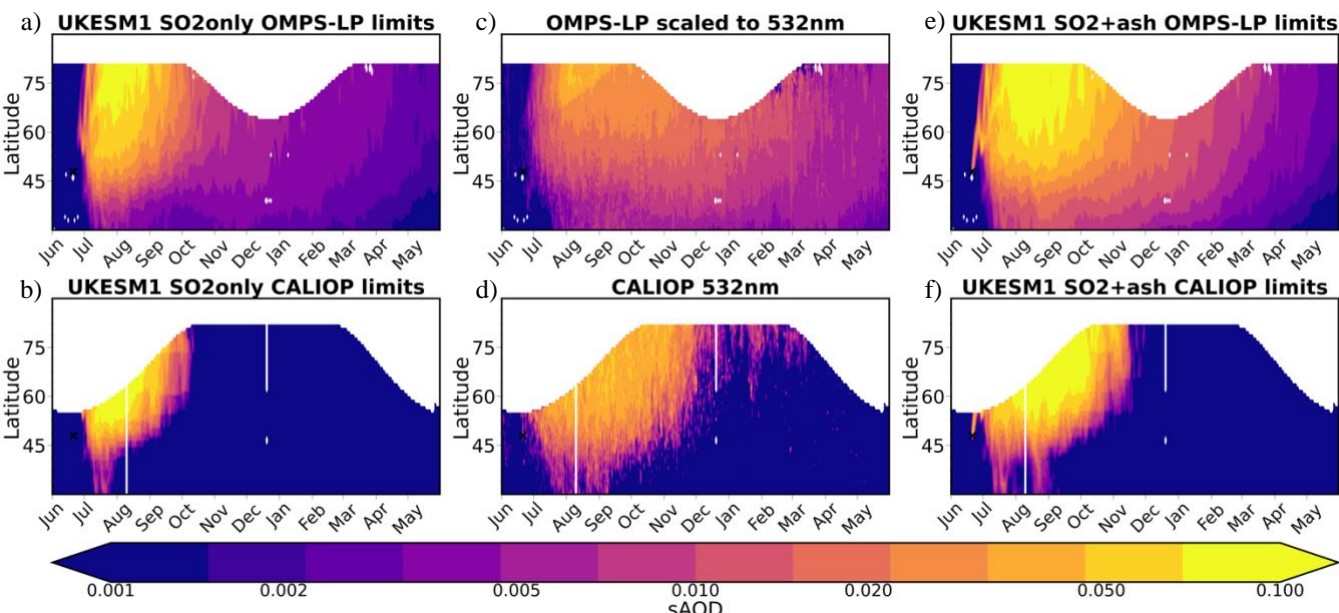

**Figure 4:** Latitude-time distribution of the zonally averaged sAOD from 30–90° N. (a) UKESM1 SO2only masked for OMPS-
LP observations, scaled from 550nm to 532nm (b) UKESM1 SO2only masked for CALIOP observations, sAOD calculated with values of aerosol extinction < 0.012 km-1, the CALIOP minimum detection limit, scaled to 532nm (c) OMPS-LP observations scaled from 869nm to 532nm (d) CALIOP observations at 532nm (e) UKESM1 SO2+ash masked for OMPS-LP observations, scaled from 550nm to 532nm (f) UKESM1 SO2+ash masked for CALIOP observations, sAOD calculated with values of aerosol extinction < 0.012 km$^{-1}$, the CALIOP minimum detection limit, scaled to 532nm. The location of Raikoke is
marked with a black cross.

Fig. 4c shows the evolution of zonal mean sAOD derived from OMPS-LP from June 2019 to May 2020. The zonal peak sAOD occurs ~1.5 months after the eruption, which is similar to the findings in Gorkavyi et al., 2021. The impact on the sAOD in OMPS-LP is still present one year later, with values of sAOD not yet returned to their pre-eruption values. This contrasts with
Fig. 4d, the same quantity derived from CALIOP, where the perturbation of sAOD is significantly reduced by December 2019 and by March/April 2020 zonal sAOD values are similar to those found pre-eruption. We can confidently attribute this

difference in aerosol lifetime to the high aerosol extinction minimum detection threshold for aerosol extinction associated with the CALIOP dataset. Figs. 4b and 4f display the UKESM1 SO2only and SO2+ash zonal mean sAOD with the CALIOP minimum retrieval limits applied where a similar distribution to that seen in the observations is modelled, indicating that the shorter aerosol lifetime observed in the CALIOP retrievals compared to OMPS-LP is due to high detection limits. As the plume disperses over time the plume becomes more diffuse and becomes undetectable by CALIOP, leading to under-detection and hence the integrated sAOD reduces much more rapidly than we see in the OMPS-LP data which does not have the same high minimum detection threshold.

In both sets of observations (Figure 4c and 4d) we can see that the enhanced stratospheric aerosol layer has been transported poleward by the Brewer-Dobson circulation with the highest sAODs found north of the eruption. The model simulations represent this transport relatively well with similar distributions to both OMPS-LP and CALIOP observations. However, in all cases the peak magnitude is over estimated, especially in the SO2+ash case. Whilst the peak sAOD in the SO2only simulation (Fig. 4a and 4b) is less of an overestimate of the observations compared to SO2+ash, it does not reproduce the evolution of the plume as well as the SO2+ash simulation (Fig. 4e and 4f) in either case. Despite the SO2+ash simulation (Fig. 4f) representing the CALIOP retrievals well it is not representative of how the plume evolves over time after becoming too diffuse for CALIOP detection limits. However, as OMPS-LP has a much lower minimum detection threshold as a dedicated stratospheric limb-profiler the decay rate of sAOD is much slower. We see a similar decay rate in the SO2+ash simulation (Fig. 4e) with comparable magnitudes to the OMPS-LP observations from December onwards. From Fig. 4 we can begin to infer that the SO2+ash simulation represents the evolution of zonal sAOD better than the SO2only case, but this inference is far from conclusive. Further comparisons to the model are made in Sect. 4.4.

### 4.3 Temporal evolution of sulfate aerosol

The CALIOP and OMPS-LP derived perturbations of sAOD from the long-term mean are presented in Figure 5. As seen in Fig. 4 the two satellite observations have different temporal evolutions. Comparing the early stages of the plume evolution it is clear that OMPS-LP does not detect the same high peak in sAOD as CALIOP, however as previously mentioned the mean CALIOP sAOD could be biased towards higher values of aerosol extinction due to fill values below the minimum retrieval limit. The CALIOP dataset shows a clear peak 60 days after the eruption with an sAOD of approximately 0.026 whereas, OMPS-LP reaches a peak sAOD of approximately 0.015 over 3 months after the eruption. Studies have suggested that limb-instruments such as OMPS-LP can fail to detect aerosol near the tropopause (e.g., Fromm et al., 2014). However, since CALIOP is a nadir viewing lidar the altitude of the plume does not significantly affect the retrieval. This could contribute to the difference we see in the initial sAOD peaks since the plume was detected at altitudes as low as 11km (Hedelt et al., 2019; Vaughan et al., 2020). The vertical profile of the plume is explored further in Section 4.6. We also see a big difference in the decay rate of sAOD. The CALIOP observations have an *e*-folding time of 84 days, in comparison to OMPS-LP which has an *e*-folding time of 220 days. As previously discussed, the difference in decay rate between CALIOP and OMPS-LP is likely

due to the different minimum detection thresholds for both satellites. Once the plume begins to dilute and become more diffuse the higher CALIOP detection threshold (0.012 km$^{-1}$) results in under-detection in comparison to OMPS-LP which has a much lower threshold (1 x 10$^{-5}$ km$^{-1}$) and is thus able to detect more of the diffuse plume.

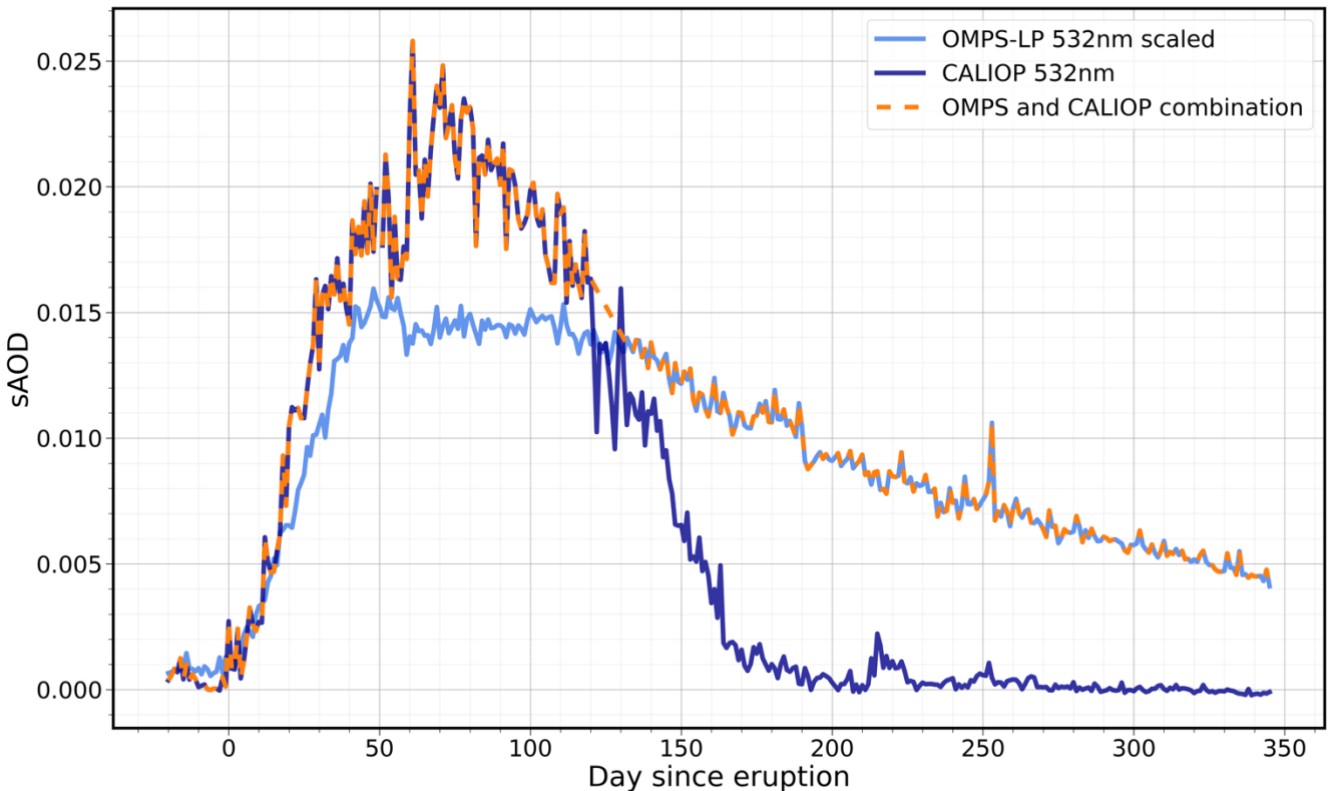

**Figure 5:** Daily perturbation in the sAOD averaged over 30–90° N as observed by OMPS-LP scaled from 510nm to 532nm
(light blue) and CALIOP at 532nm (dark blue). The orange dashed line represents the combined OMPS-LP and CALIOP dataset at 532nm. We remove the long-term background sAOD derived from OMPS-LP (0.0041) and CALIOP (0.0003) for the years 2013–2018 from those for 2019 to provide a stratospheric perturbation for the observations.

We have created a combined dataset which includes aerosol extinction data from both CALIOP and OMPS-LP, seen in Fig.
5. The combined dataset utilises the area averaged (30 - 90°N) CALIOP sAOD data for the first 4 months after the eruption before it is linearly interpolated over the region in which the two datasets cross over and then employs OMPS-LP data for the remaining months. This new dataset has an *e*-folding time of 145 days, and we believe that it is more physically reasonable than using a single observational dataset due to the data constraints outlined above. To confirm qualitatively that using the OMPS-LP data is more appropriate than CALIOP in the later months after the eruption we use in-situ ground-based data to
provide an alternative comparison to these satellite datasets. We utilize the AERONET Level 2 AOD retrievals measured at

550nm and scale them to 532nm for comparison with the satellite observations. To calculate the satellite retrievals at MLO an area average is taken across multiple grid boxes encompassing the MLO. Note that AERONET retrievals of sAOD are not a point measurement as they are a function of the solar zenith angle. For solar zenith angles of 60-80 degrees and assuming that any aerosol is located in the lowest 20km above the observatory, aerosol within a 35-115km radius of the Mauna Loa observatory is included in the observations. The same area is used to calculate the average monthly SO2only and SO2+ash perturbations.

| | AERONET | OMPS-LP | UKESM1 SO2only | UKESM1 SO2+ash |
|---|---|---|---|---|
| August | $5.89 \times 10^{-3}$ | **$2.16 \times 10^{-3}$** | $2.78 \times 10^{-3}$ | $4.27 \times 10^{-3}$ |
| September | **$12.12 \times 10^{-3}$** | **$7.07 \times 10^{-3}$** | $3.03 \times 10^{-3}$ | $8.46 \times 10^{-3}$ |
| October | **$9.78 \times 10^{-3}$** | **$6.41 \times 10^{-3}$** | $3.33 \times 10^{-3}$ | $7.12 \times 10^{-3}$ |
| November | **$4.70 \times 10^{-3}$** | **$4.89 \times 10^{-3}$** | $2.91 \times 10^{-3}$ | $5.40 \times 10^{-3}$ |
| December | $1.05 \times 10^{-3}$ | **$3.81 \times 10^{-3}$** | $2.12 \times 10^{-3}$ | $3.96 \times 10^{-3}$ |
| Average | $6.71 \times 10^{-3}$ | $4.87 \times 10^{-3}$ | $2.83 \times 10^{-3}$ | $5.84 \times 10^{-3}$ |

**Table 3:** Perturbation of AOD from the long-term mean retrieved from the Mauna Loa Observatory AERONET site scaled to 532nm. OMPS-LP, UKESM1 SO2only and SO2+ash sAOD perturbation calculated across an area encompassing the MLO (19–20° N, 152–156° W). Observations highlighted with bold text are statistically significantly greater than the long-term mean at 95% confidence level. Negative values of AOD are a result of calculating the perturbation from the long-term mean (2013–2018).

Average monthly perturbations from the long-term mean (or control simulation) for the Mauna Loa observatory, are presented in Table 3. The AERONET retrievals are statistically significant at the 5% level from the long-term mean from September to November and increase to a peak AOD of $12.12 \times 10^{-3}$ in September, over 2 months after the eruption. The OMPS-LP retrievals and SO2+ash model show a similar pattern with peak AODs in September of $7.07 \times 10^{-3}$ (OMPS-LP) and $8.46 \times 10^{-3}$ (SO2+ash). The SO2only simulation follows a similar pattern of increased AOD between August and November, however the magnitude of AOD is much smaller than the AERONET and OMPS-LP observations. OMPS-LP retrievals are also significantly greater than the long-term mean from August through until December. CALIOP however does not appear to detect any statistically significant perturbation to the sAOD, with values an order of $10^2$ smaller than those observed by AERONET and OMPS-LP. This is most likely due to aerosol loadings falling below the minimum detection threshold and the plume becoming more diffuse at this latitude. We also calculate the model average perturbed AOD for this region from August

to December, presented in Table 3. The SO2+ash average AOD, 5.84 x 10$^{-3}$, agrees well with the AERONET and OMPS-LP observations, 6.71 x 10$^{-3}$ and 4.87 x 10$^{-3}$ respectively. However, the SO2only average AOD is much smaller suggesting that the SO2+ash simulation validates better against this specific set of observations.

MLO is located at 19.5°N, around 30° south of Raikoke. Due to its proximity to the 5°S Ulawun eruption, there is the potential for this eruption to influence observational data. Model simulations were run without Ulawun emissions (discussed further in Sect. 4.7) and a negligible influence on the AOD in the MLO region over this time period was observed. While there may indeed be an influence from Ulawan on the AODs determined at Mauna Loa in the observational record, this modelling and the observed timing of the statistically significant AOD perturbations suggest that any influence is small. As we noted in Fig. 4 most of the aerosol plume travelled poleward via the Brewer-Dobson circulation, however there was some southern transport seen in both satellite observations in late July and August. We can observe this further in Figure 6, the monthly average sAOD observed by OMPS-LP and CALIOP.

A similar analysis performed by Haywood et al. (2010) for the Sarychev eruption reveals 550nm AOD perturbations of +0.010 and +0.008 for July and August. Given that the Sarychev and Raikoke eruptions occurred within a few calendar days of each other but in different years, it appears that the equatorward transport of aerosol for the Sarychev eruption was quicker than for the Raikoke eruption. This conclusion holds despite any potential influence from the eruption of Ulawun and highlights that considerable differences in transport can occur for volcanic eruptions that are ostensibly very similar in terms of latitude, timing, injection amount and vertical distribution (Jones et al., 2016).

Figure 6 shows the monthly geographic evolution of the sAOD in the Northern Hemisphere. MLO is highlighted on the first plot by a red cross. From this plot we can see that transport to lower latitudes does not occur until August, however the CALIOP retrievals are much more diffuse than those observed in OMPS-LP. We observe high values of sAOD at high latitudes with peaks across Greenland and Northern Canada. Despite the missing data in the CALIOP observations we can still see a reasonable spatial agreement in the sAOD during the first few months. In October an interesting feature is seen in the OMPS-LP data where a band of enhanced sAOD is observed between 0 – 15°N. This might be attributed to the second Ulawun (5.05°S, 151°E) eruption on 3$^{rd}$ August which, owing to the latitude and altitude of the eruption is likely to be confined by the so-called "tropical pipe" between approximately 15°S-15°N (e.g. Plumb, 1997), although some leakage to higher latitudes might be expected over time. From August onwards the stratospheric aerosol layer south of the equator has been shown to become enhanced. Kloss et al., (2021) estimate that the aerosol from the Ulawun eruption circled the Earth in the tropics within one month. During October and November the sAOD in the tropical stratosphere becomes increasingly enhanced, which is likely due to the influence of Ulawun. Due to the influence of Ulawun this study uses area averages from 30 – 90°N to ensure analysis focuses solely on the impact of the Raikoke eruption. Despite the potential influence of Ulawun on MLO observations, we can nevertheless conclude that combining the initial CALIOP peak and the latter half of the OMPS-LP data is the most

appropriate representation of the plume evolution since OMPS-LP does not observe the initial peak while CALIOP detection limits lead to non-detection of aerosol once it has become diffuse hence apparently reducing the observed lifetime and e-folding time of the sulfate aerosol.

510

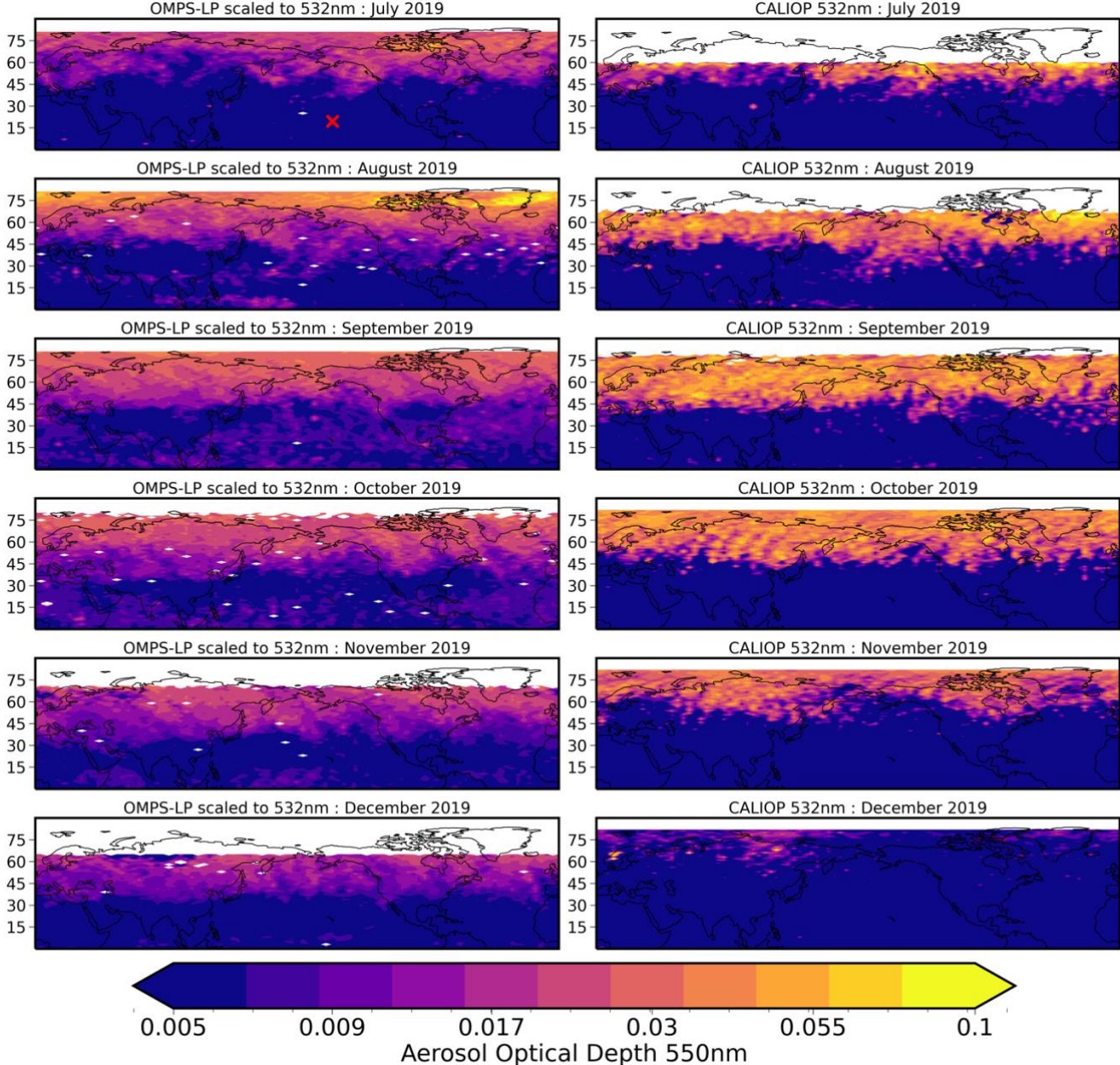

**Figure 6:** Monthly geographic evolution of the Northern Hemisphere sAOD from July 2019 to December 2019 derived from OMPS-LP Retrieved Aerosol Extinction (left) and CALIOP Aerosol Extinction Profile (right). We remove the long-term

background sAOD derived from OMPS-LP and CALIOP for the years 2013–2018 from those for 2019 to provide a stratospheric perturbation for the observations.

## 4.4 Model comparison

To make consistent and accurate comparisons between the combined satellite dataset and the model simulations it is appropriate to implement the same method used in Fig. 4 and apply minimum detection and spatial limits to the model data. Figure 7 compares the combined OMPS-LP and CALIOP dataset to the two model simulations both with (solid lines) and without (dashed lines) the respective observational limits applied. For the first 4 months after the eruption the combined dataset uses the CALIOP observations, therefore we use the CALIOP minimum detection threshold (0.012 km$^{-1}$) to filter out aerosol extinction values in the model simulations before calculating the sAOD for comparison. After this we apply the limits of the OMPS-LP data to the model simulations in a similar fashion since OMPS-LP is used in the combined observational dataset for the following months. For both SO2only and SO2+ash we only include model data where observational data were available. When the observational limits were applied, no linear interpolation across the location where the combined dataset switches from CALIOP to OMPS-LP was applied to the model simulations owing to the considerable difference in sAOD across the area of interpolation.

Figure 7 clearly demonstrates significant differences between the sulfate aerosol evolution in the SO2only and SO2+ash simulations. Considering first the SO2only simulations, we note that the timing of the peak is much earlier in SO2only compared to the observations. The combined observational dataset peaks at approximately 0.026 around 2 months after the eruption and has an *e*-folding time of 145 days. For the SO2only simulations when including the observational limits, which is considered the most appropriate method of comparison, the peak sAOD is much greater (0.033) and earlier than observations. We observe a similarly fast decrease in sAOD when observational limits are applied to SO2only (*e*-folding time of 45 days), whereby the sAOD drops to values close to zero at day 110 before an increase at approximately day 125 after the eruption. This abrupt change is an artifact of combining CALIOP and OMPS-LP limits to the SO2only simulation to best compare against the combined observational dataset.

Now considering the SO2+ash simulations. In contrast to the SO2only simulation, there is a large difference in the first 50 days after the eruption between the SO2+ash model with and without limits. When observational limits are applied (as recommended), the peak sAOD (0.042), whilst still an overestimate, occurs 60 days after the eruption which is more consistent with the combined observational dataset. The delay in the peak of sAOD when observational limits are applied can be attributed in part to the missing data at high latitudes during the polar summer since we can still identify this delay in the peak sAOD when we only apply spatial CALIOP limits to SO2+ash. We can see in both Fig. 4 and 6 the sulfate aerosol travels poleward due to the Brewer-Dobson circulation and we would expect the resulting sAOD to be greatest within the first few months after

the eruption. However, since the months following the eruption are the Northern Hemisphere summer, there are no CALIOP night retrievals at high latitudes, contributing to the appearance of a delayed sAOD peak.

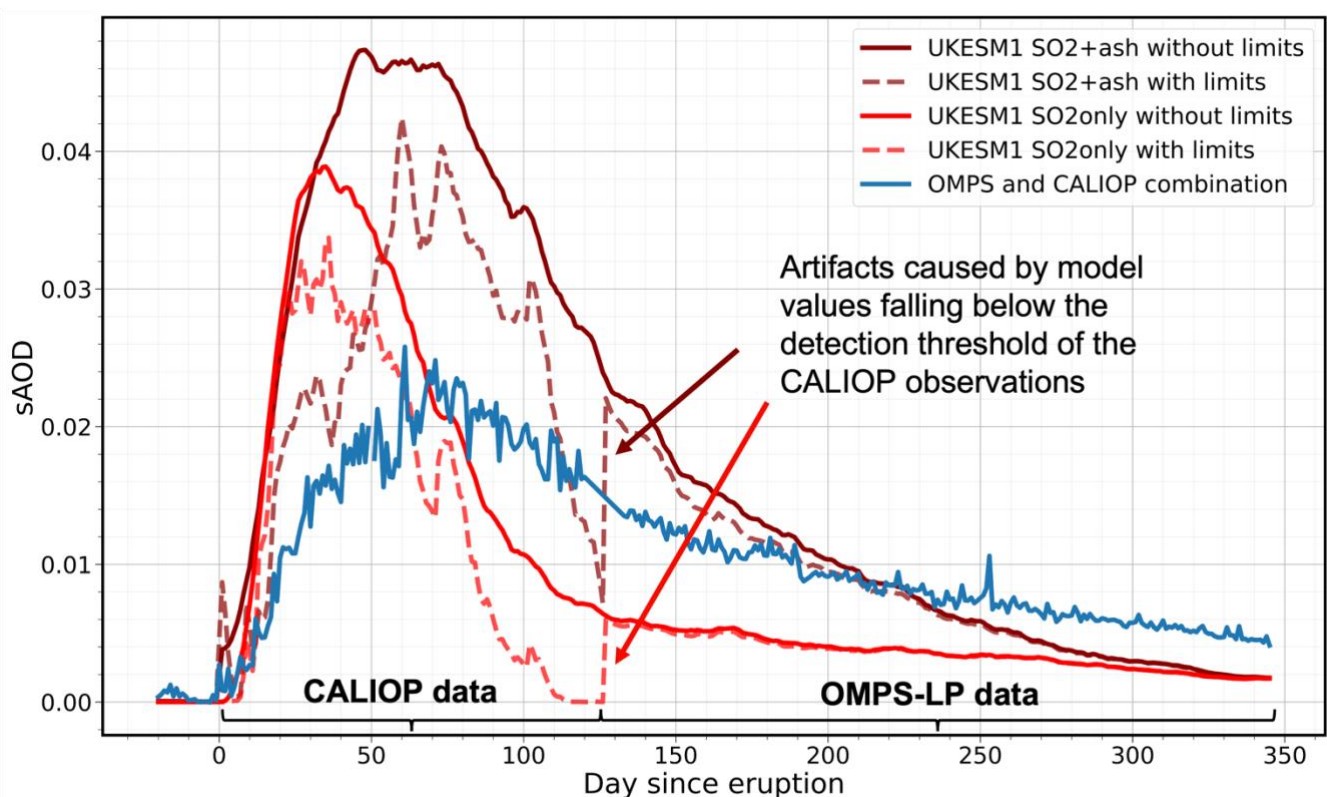

**Figure 7:** Daily perturbation of sAOD at 532nm averaged over 30–90° N. Daily OMPS-LP and CALIOP combined dataset sAOD (blue), UKESM1 SO2only with observational limits applied (red) and without limits applied (red dashed) and UKESM1 SO2+ash with observational limits applied (dark red) and without limits applied (dark red dashed). We remove the long-term background sAOD derived from OMPS-LP and CALIOP for the years 2013–2018 from those for 2019 to provide a stratospheric perturbation for the observations. Similarly, we remove the impacts of background stratospheric aerosol from the model simulations by subtracting the sAOD from the CNTL simulation from those for SO2only and SO2+ash.

555

When observational limits are applied to SO2+ash we observe a decline in model sAOD after the peak owing to transfer from the stratosphere to the troposphere with an *e*-folding time of 90 days, almost 2 months faster than the observed data. The initial steep decline in SO2+ash is due to the CALIOP limits imposed upon the model. At approximately day 135 after the eruption there is a sharp increase in modelled sAOD. This is an artifact of combining SO2+ash with CALIOP limits and SO2+ash with OMPS-LP limits. The SO2+ash simulation with CALIOP limits applied has a much faster decay rate with an *e*-folding time of 44 days, whereas the SO2+ash simulation with OMPS-LP limits has a much longer *e*-folding time of 101 days. Whilst

SO2+ash is not perfect at recreating the sAOD evolution, we can see that by applying the observational limits it starts to become apparent that SO2+ash provides a better comparison against observations than SO2only, although some considerable differences still exist. In the following two sections we explore changes to the stratospheric aerosol Ångström Exponent and the vertical profile evolution of the aerosol to further examine the consistency of the SO2only and SO2+ash simulations against observations.

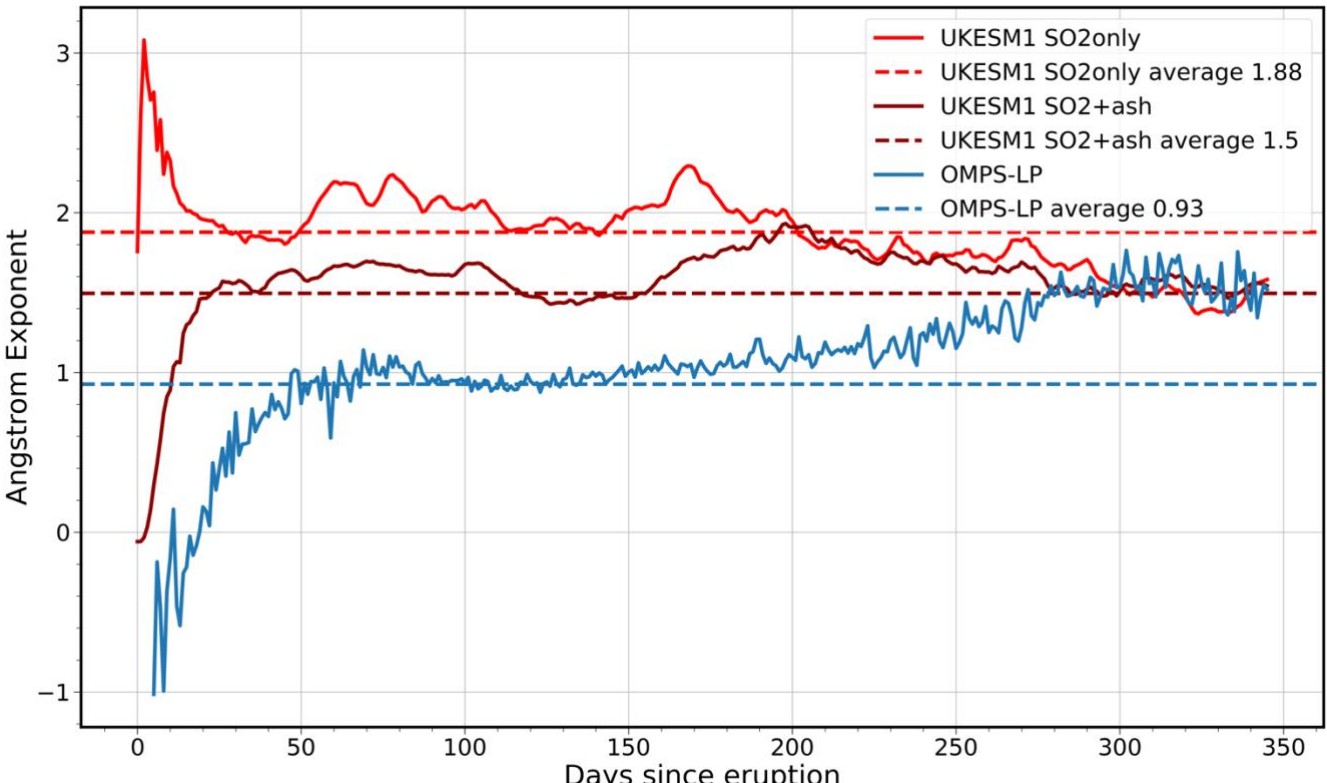

**Figure 8:** Daily evolution of Ångström Exponent for OMPS-LP (blue), UKESM1 SO2only (red) and UKESM1 SO2+ash (dark red). Calculated using area weighted sAOD between 30–90° N (same as Fig. 7) using 510nm and 869nm wavelengths.

**4.5 Analysis of stratospheric aerosol Ångström Exponents**

Without detailed in-situ measurements (e.g. Jégou et al., 2013) it is not possible to know with a high level of accuracy, the detailed size distribution of the volcanic aerosols. However, the wavelength dependence can be used to calculate the Ångström Exponent, since it is often used as an indicator of aerosol particle size. For measurements of optical depth and at wavelengths and the Ångström Exponent is given by Eq. (1):

$$\alpha = -\log\frac{\tau_{\lambda_1}}{\tau_{\lambda_2}} / \log\frac{\lambda_1}{\lambda_2}, \tag{1}$$

The OMPS-LP aerosol extinction measurements are provided at wavelengths ranging between 510 – 997nm, presenting the opportunity to calculate the sAOD and analyse the evolution of the Ångström Exponent and consequently variations in the aerosol size distribution after the eruption. Figure 8 shows the daily Ångström Exponent (AE) for OMPS-LP, SO2only and SO2+ash calculated using the area averaged sAOD at wavelengths 510nm and 869nm between 30 – 90°N.


The observations show that the AE is initially close to between -1 and 0 indicating that large particles were observed immediately after the eruption, and therefore contains a significant amount of volcanic ash. After around 50 days after the eruption, the AE increases to approximately 1 for a period of over 3 months, this could suggest that the largest particles had dropped out and smaller particles remained. As the time after the eruption increases, the AE increases to a maximum value of

~1.6. Kloss et al., 2021 estimate a pristine average AE of 1.7 using background sAODs which suggests that approximately 300 days after the eruption the observations have returned to pre-eruption AE average.

Both model simulations converge on an AE of around 1.6 after 300 days, however they both display starkly contrasting behaviour immediately after the eruption. In the SO2only scenario the initial AE is very high (up to around 3) indicating

smaller particles and decreases with time. The behaviour of the AE is therefore the opposite of what is observed.

In comparison, during the first 10 days after the eruption, the SO2+ash scenario shows an AE of around zero initially owing to the presence of ash, which then increases as the ash falls out from the atmosphere. This behaviour is much more similar to what we see in the observations as compared to SO2only, confirming that our SO2+ash simulations are in better agreement

with observations. However, the agreement is far from perfect with the model AE increasing much faster than what we see in the observations, it then converges with the SO2only scenario approximately 200 days after the eruption. Considering an internal mixture between ash and sulfate in the model could potentially resolve the differences between the observations and the model AE. However, the slower increase in the observed AE could also indicate an influence from pumice or soot particles from forest fires. Despite these differences, this figure again suggests that the SO2+ash scenario is better at representing the

observations compared to the SO2only case. Further discussion of this is provided in Sect. 5.

## 4.6 Aerosol extinction vertical profile analysis

Figure 9 shows the vertical evolution of the CALIOP derived aerosol extinction coefficient with monthly averages and standard deviation from July until December. We also include both model simulations, SO2only (dashed red line) and SO2+ash (solid red line) for comparison. The observed monthly tropopause height (mean and standard deviation) is also included to highlight

tropospheric and stratospheric altitudes. Both observations and model simulations are averaged over 30 – 90°N, not including those latitudes where CALIOP night retrievals are unable to retrieve data due to polar summer. Since CALIOP aerosol extinction retrievals have a minimum detection threshold of 0.012 km$^{-1}$ (Toth et al., 2018) this limit has also been applied to the SO2only and SO2+ash data for a more consistent comparison.

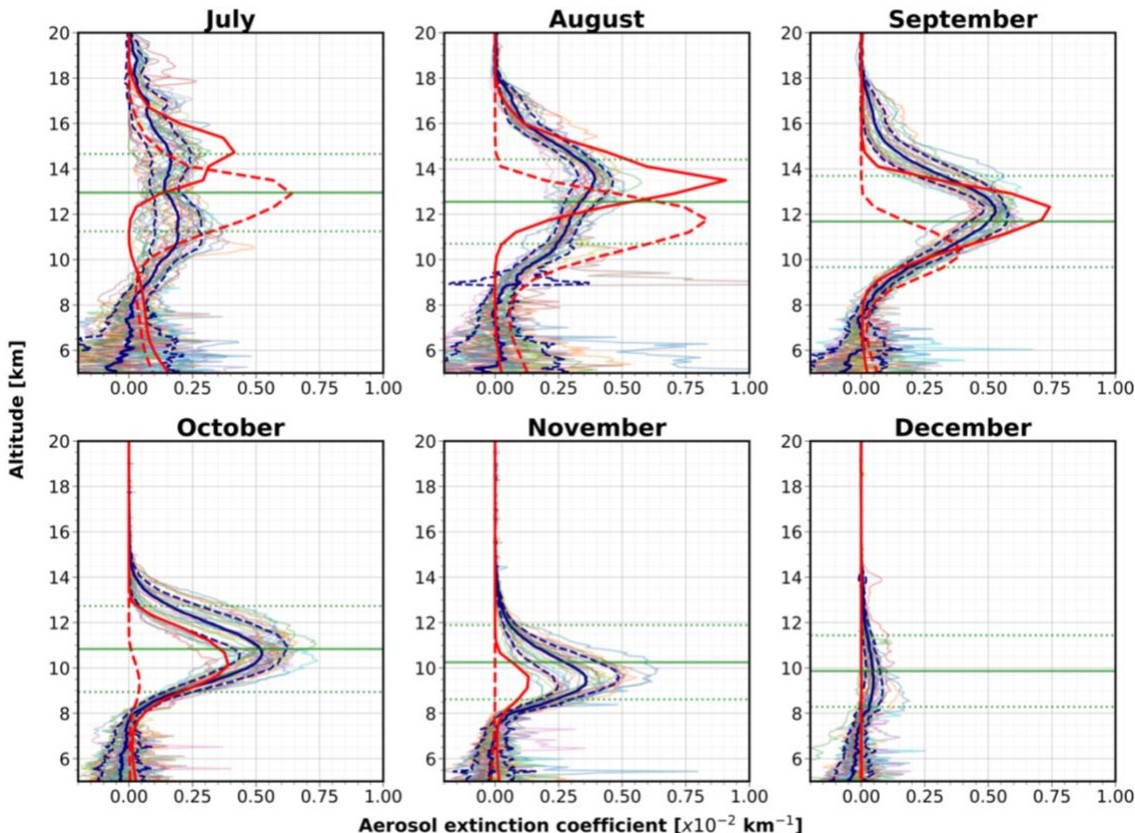

**Figure 9:** Aerosol extinction coefficient vertical profile averaged longitudinally and over 30–90° N. Averaged daily CALIOP aerosol extinction coefficient vertical profiles (night retrievals only, fainter lines) with monthly average (bold blue) and monthly average plus/minus standard deviation (solid dashed black lines). UKESM1 SO2+ash (solid bold red) and SO2 only (dashed bold red) simulations with imposed CALIOP minimum retrieval limits and mask. Average tropopause height is shown by the horizontal green line and the average tropopause height +/- one standard deviation for 2019 is also shown in green dotted lines.

In the CALIOP observations we can see that initially after the eruption there are two peaks at ~11km and ~14.5km similar to the two injection altitudes used to initialise the UKESM1 simulations. The observations then form a singular peak above the average tropopause at ~14km which increases in magnitude to a peak of 5.3 x $10^{-3}$ km$^{-1}$ between September and October. In comparison, the two model simulations peak at 8.3 x $10^{-3}$ km$^{-1}$ for SO2only at ~12km and 9.1 x $^{-3}$ km$^{-1}$ for SO2+ash at ~13.5km. We notice a considerable difference in the altitude of the plume between the two model simulations. In July the peak aerosol extinction for SO2only is at approximately 13km, a similar altitude to the average tropopause height. In contrast to this, the SO2+ash plume has a peak aerosol extinction at ~15km. This initial difference in height results in a significant impact on the lifetime of the aerosol. To explore the impact of the altitude of the plume further, we look at Figure 10 which displays the

aerosol extinction coefficient as a function of latitude and altitude. This avoids averaging over a widely varying tropopause height and allows us to examine the altitude of the plume in the models and observations against the altitude of the tropopause and how it might affect the aerosol lifetime.

As per the analysis shown in Fig. 9, we impose the CALIOP detection limits onto the model simulations and only include

latitudes in which CALIOP data is available. The average monthly tropopause height is included and shown by the black line. The initial difference between the plume altitude of SO2only and SO2+ash is striking. Already in July we observe some aerosol in the midlatitudes below the monthly tropopause height, with a substantial portion below the tropopause in August. However, we do not see this in the SO2+ash model until September and October. The CALIOP observations reveal the poleward transport of the aerosol via the Brewer-Dobson circulation from July through until September. This is reproduced reasonably well in the

SO2+ash model, however we do not observe this in the SO2only simulation. Similarly, the magnitude and spatial distribution of the aerosol in October is well modelled by SO2+ash compared to SO2only in which only a negligible aerosol extinction coefficient is found.

Due to more removal processes in the troposphere the altitude of the plume in relation to the tropopause height can impact the

decay rate and lifetime of the aerosol. Since both model scenarios were initialised with the same $SO_2$ injection altitudes the difference between plume altitudes in July must be due to the self-lofting effect from the ash. Muser et al., 2020 had documented this for the Raikoke plume and noted that the maximum cloud-top height rose more than 6km within the first few days after the eruption. Whilst the SO2+ash scenario overestimates the magnitude of the aerosol extinction coefficient in the first few months, it does represent the altitude and lifetime of the aerosol plume well. The impact of the self-lofting effect is

also seen in Fig. S2, which displays the stratospheric aerosol optical depth of SO2+ash and SO2only, alongside the sAOD derived from the CLASSIC dust emission and SO2+ash minus the sAOD from dust. We can see here that the impact from the ash itself is relatively negligible and therefore it must be the impact the ash has on the sulfate aerosol which drives the changes seen in sAOD.

The vertical profile analysis of the CALIOP aerosol extinction coefficient indicates how close the volcanic plume was to the tropopause in the first month or so after the eruption. This could explain why the OMPS-LP dataset missed the initial sAOD peak since limb-instruments can fail to detect aerosol near the tropopause (e.g. Fromm et al., 2014). Analysis of the aerosol size evolution and vertical profile in the model simulations have confirmed that the UKESM1 SO2+ash simulation is more consistent with the observations. Figs. 8, 9 and 10 illustrate that SO2+ash provides a much better representation of the aerosol

size and lifetime and therefore resulting climatic impact than the SO2only scenario.

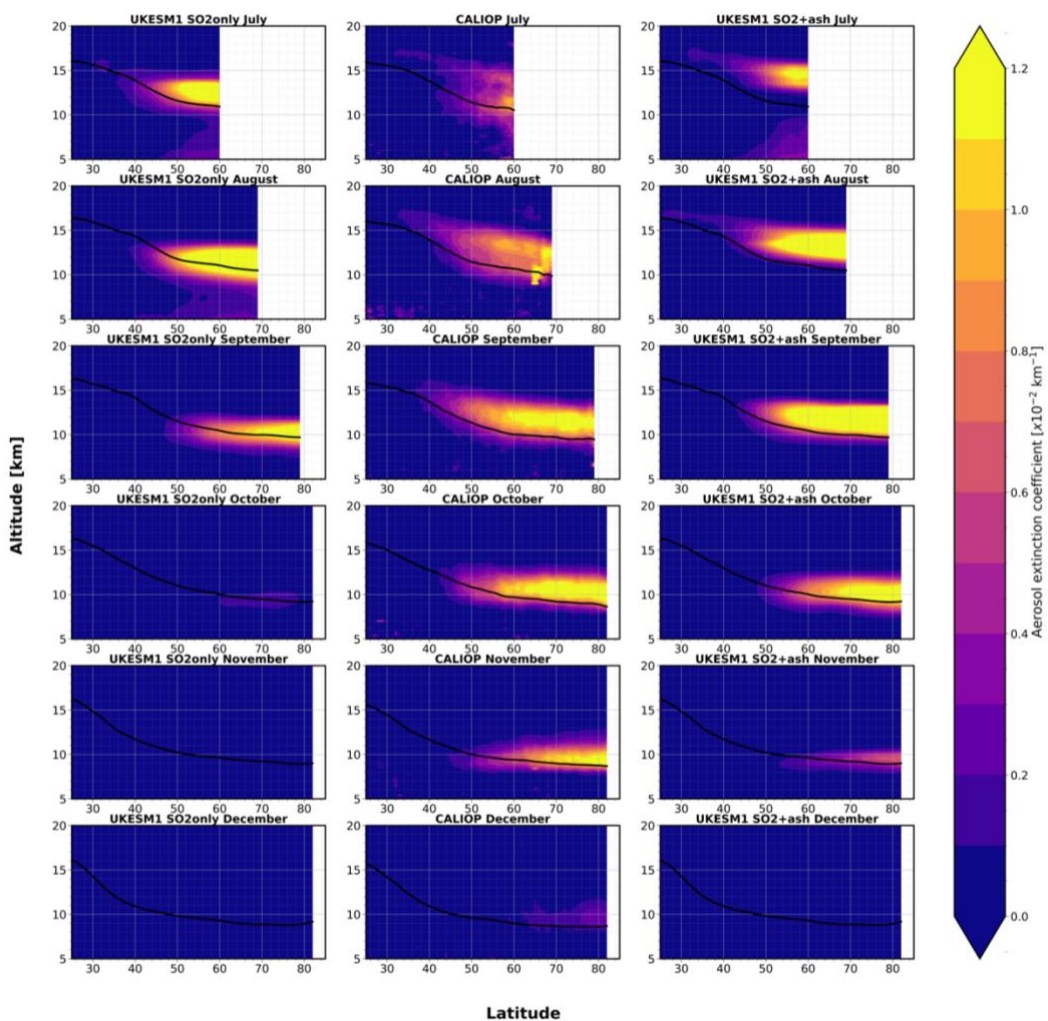

**Figure 10:** Aerosol extinction coefficient vertical profile averaged longitudinally. Averaged monthly CALIOP (centre) aerosol extinction coefficient vertical profiles (night retrievals only) with monthly average tropopause height (solid black). UKESM1 SO2 only (left) and SO2+ash (right) simulations with imposed CALIOP minimum retrieval limits and mask.

### 4.7 Radiative forcing

Haywood et al., 2010 ran simulations of the Sarychev eruption over a three-month period June-August 2009 and explored the difference between the aerosol optical depth and radiative forcing due to anthropogenic aerosol and the Sarychev Peak stratospheric aerosol plume. They found that in some regions of the NH the AOD was of comparable magnitude or greater than the AOD from anthropogenic emissions. We estimate the impact of the Raikoke eruption on the approximate radiative forcing exerted by the volcanic aerosol and compare this to the Sarychev Peak eruption. In Figure 11 we have recreated the

Sarychev Peak AOD and radiative forcing plots from Haywood et al., 2010 Figure 12 using data from HadGEM2 and compare them against our UKESM1 Raikoke simulations, eliminating the impacts of the Ulawun eruption. As in the results presented by Haywood et al. (2010), our UKESM1 simulations are nudged to reanalysis data, but the evolution of the control and experiment simulations differ slightly and result in some differences in the cloud and fields. Due to this, the all-sky radiative forcing cannot be accurately determined from the difference in the top of the atmosphere shortwave upwelling radiation with and without aerosols. The all-sky shortwave radiative forcing can be approximated for purely scattering aerosol, following the calculation used by Haywood et al., (2010), using the following equation, Eq. (2):

$$Radiative\ forcing \approx (SW_{CTL} - SW_{EXP}) * (1 - A_c), \tag{2}$$

where $SW$ is the clear-sky outgoing shortwave radiation (Wm$^{-2}$) at the top of the atmosphere (TOA) for the control (CTL) and experiment (EXP) and $A_c$ is the cloud fraction (Haywood et al., 1997). This assumes that the contribution from longwave radiation and the radiative forcing from cloudy areas is negligible. It has been shown through radiative transfer calculations that for conservative scattering from sulfate aerosols for optically thick cloud that this is a reasonable assumption (e.g. Haywood and Shine, 1997).

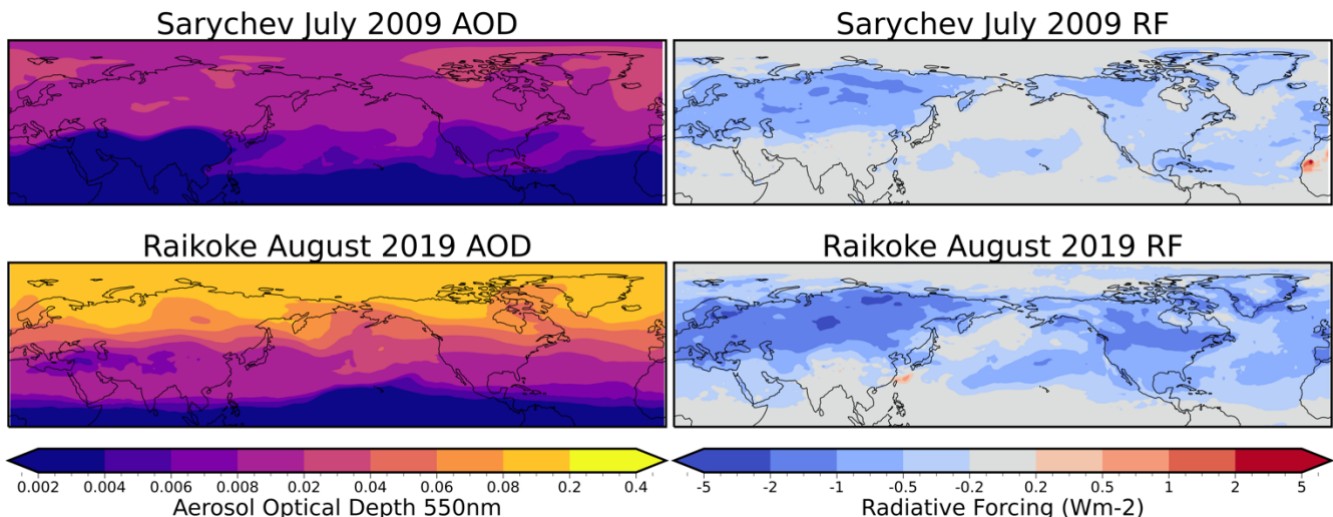

**Figure 11:** The sAOD for July at 550nm for a) Sarychev Peak derived from HadGEM2 and b) Raikoke derived from UKESM1. The cloud-free radiative forcing at the top of the atmosphere (Wm$^{-2}$) for c) Sarychev Peak derived from HadGEM2 and d) Raikoke derived from UKESM1.

Figure 11 shows the months in which the different eruptions peak in both AOD and radiative forcing. In order to make global comparisons between the Sarychev Peak eruption and Raikoke, we perform a UKESM1 simulation in which we do not include the Ulawun eruption. We see in both plots that the Raikoke eruption had a greater impact on the global AOD and resulted in a

larger cloud-free radiative forcing at its peak. After the Sarychev Peak eruption the NH average AOD peaked in July at approximately 0.01, this is in comparison to Raikoke, where the peak NH averaged AOD occurred in August at approximately 0.03. The global and NH radiative forcing impacts from both eruptions from June through until August are shown in Table 4.

| | Raikoke SO2only | **Raikoke SO2+ash** | Sarychev Peak |
|---|---|---|---|
| June | -0.001 | **-0.004** | -0.03 |
| July | -0.26 | **-0.22** | -0.13 |
| August | -0.29 | **-0.25** | -0.07 |
| September | -0.13 | **-0.19** | N/A |
| October | -0.05 | **-0.14** | N/A |
| November | -0.02 | **-0.07** | N/A |

**Table 4:** The cloud-free global mean radiative forcing (in $Wm^{-2}$) at the top of the atmosphere for Sarychev Peak (after
Haywood et al., 2010) derived from HadGEM2 and for Raikoke SO2only and SO2+ash simulations derived from UKESM1. N/A indicates not available.

Rather counter-intuitively, the cloud-free radiative forcing is some 20% stronger for the SO2only simulations than for the SO2+ash simulations. The explanation for this result is that, as in the case for clouds, the evolution of the aerosol differs
slightly between the SO2only and the SO2+ash simulations. This results in sAODs that are some 20% larger over areas of central Eurasia in July and August, which are predominantly cloud-free in the model at that time of year. This results in cloud-free radiative forcings that are a factor of 1.2 greater for the SO2only simulations than the SO2+ash simulations. However, far greater fractional differences of up to a factor of 3.5 are evident in subsequent months owing to the general reduction in the sAOD documented in previous sections.


While the cloud-free radiative forcing from June cannot be directly compared owing to differences in timing of the eruptions (Sarychev eruption initiated 15th June in Haywood et al., (2010), but 21st June in this study), the global mean cloud-free radiative forcing from the Raikoke eruption in comparison to the Sarychev Peak eruption are substantially greater in July and August. At their peaks, the NH cloud-free radiative forcing of Raikoke is -0.48 $Wm^{-2}$, almost twice as strong as that of Sarychev
Peak (-0.24 $Wm^{-2}$). This is due to a combination of factors including but not limited to i) the base-state model (UKESM1 versus HadGEM2), ii) the aerosol modelling scheme (UKCA versus CLASSIC), iii) differences in assumed vertical emission profile (80% at 13-15km versus evenly distributed between 11-15km), iv) differing meteorological conditions which have been shown to significantly impact aerosol distribution, lifetime, and hence radiative forcing, particularly for emissions close to the tropopause (Jones et al., 2016). Note that the inclusion of volcanic ash does not appear to be a significant cause of the

increased peak radiative forcing compared to the Sarychev analysis; the inclusion of ash appears to have a greater impact on the longevity of the radiative forcing through enhanced stratospheric aerosol lifetime. To fully evaluate differences across model formulations would require re-runs of the model for the Sarychev case with UKESM1, which is beyond the scope of this study.

## 5. Summary and conclusion

This study provides an analysis of satellite and ground-based observational data to determine if including ash in model simulations of volcanic eruptions can more accurately represent aerosol size, geographic distribution, and the evolution of volcanic plumes. There are substantial differences between observations due to different observational limitations. Whilst these were difficult to reconcile, we were able to apply numerous observational thresholds to UKESM1 to assess the ability of the model to replicate observations. Using multiple remote sensing methods we were able to validate the transport of the $SO_2$

plume, the evolution of the sulfate aerosol and associated radiative impacts modelled by the UKESM1 nudged to ERA5 reanalysis data.

The Raikoke eruption was the largest volcanic injection of $SO_2$ into the stratosphere since the OMPS satellite was launched and was well observed by OMPS-NM and OMPS-LP. This revealed that the plume became trapped within a cyclonic

circulation for several days across Eastern Russia and Alaska before travelling eastwards across North America and the North Atlantic. This agrees with other satellite observations used in previous studies (e.g., Kloss et al., 2021, Vaughan et al., 2020). When nudged to ERA5 reanalysis data the UKESM1 SO2only and SO2+ash simulations were able to represent the position of the main features of the plume evolution, including the cyclonic circulation. Similarly to other models (e.g. Haywood et al., 2010; de Leeuw et al., 2021) the model simulations produce a more diffuse $SO_2$ plume compared to observations due to a

combination of model uncertainty and observational limits.

The distribution of the sulfate aerosol plume was examined using two satellite observations with comparisons to the UKESM1 model simulations. The zonal evolution observed by CALIOP, and OMPS-LP followed a similar geographic evolution to the Sarychev Peak eruption in June 2009 (Haywood et al., 2010) which was to be expected since they are neighbouring volcanoes

and the altitude and injection magnitude of Sarychev Peak (11 – 15km, $1.2 \pm 0.2$ Tg) were similar to that of Raikoke (10 – 15km, $1.5 \pm 0.2$ Tg). Due to differences in minimum detection thresholds it was observed that the CALIOP data did not represent the long-term evolution of the sulfate aerosol plume well and similarly to other limb-instruments, OMPS-LP can fail to detect aerosol near the tropopause at the beginning of an eruption. Hence, we created a combined dataset consisting of both CALIOP and OMPS-LP aerosol extinction data. Data from the Mauna Loa Observatory provided additional corroborative

evidence that the OMPS-LP observations were more appropriate after the plume had dispersed, with CALIOP observations showing no significant increase in sAOD for the eruption year. Model simulations differed in the MLO region, with the

SO2+ash case presenting similar values of sAOD to both the OMPS-LP and AERONET observations. In contrast, the SO2only simulation revealed values of sAOD much lower than those seen in the observations and SO2+ash case.

We then studied the observed Ångström Exponent utilising the OMPS-LP 869nm and 510nm wavelength aerosol extinction coefficient retrievals. Throughout the first 50 days after the eruption the AE was less than one, indicating the presence of large aerosol particles. Comparing the observations to both the SO2only and SO2+ash UKESM1 scenario demonstrates that whilst both models do not capture the observations completely, the SO2+ash scenario better represents the change in aerosol size after the eruption. It is also an indicator that the model removes the ash much faster than we see in the observations. Zhu et al.,

2020 proposed that after the 2014 Mount Kelud eruption the volcanic aerosol layer was primarily composed of low density, super-micron sized ash. Most model simulations assume volcanic ash particles are denser than the pumice-like particles observed in Zhu et al. (2020) and hence the ash particles in model simulations fall out more quickly. Supplementary figure 2 highlights the relatively small and short impact the ash in the model has on the stratospheric aerosol optical depth itself. Instead we see in Figs. 9 and 10 how the ash produces a self-lofting effect onto the sulfate aerosol, extending the stratospheric lifetime

of the aerosol. This could explain the difference between the observations and the SO2+ash model case. Obviously, there are limitations to our assumption that the ash is externally mixed with the sulfate aerosol as in reality there will be varying degrees of internal and external mixture.

It is possible that the observations were also influenced by the Canadian and Siberian wildfires which occurred in the summer

of 2019. A few days before the Raikoke eruption (17th June 2019), wildfires in Alberta, Canada produced biomass burning aerosols and formed pyrocumulus clouds which entered the lower stratosphere. By late June the resulting aerosol layers arrived over the UK and Europe, this was only a few days before aerosols originating from the Raikoke plume were observed by UK lidars (Osborne et al., 2022). The Siberian wildfires were extreme and lasted from 19th July to 14th August 2019 (between 28-54 days after the eruption; Johnson et al., 2019) injecting wildfire smoke into the troposphere near to the Arctic region.

Ohneiser et al., (2021) and Ansmann et al, (2021) suggested that the influence from the Siberian wildfire smoke increased the aerosol optical depth across the Arctic region. However, this finding is challenged by Boone et al., (2022). Whilst there might have been some influence on the aerosol optical properties and distribution from these events, our simulations suggest that the SO2+ash model provides a reasonable representation of the Raikoke eruption without including the Siberian wildfires in the model simulation. Our work certainly shows that using the results from a single climate model simulation of Sarychev to infer

aerosol optical depths for other volcanic eruptions such as Raikoke is inadvisable. Further work appears necessary to elucidate whether biomass burning smoke aerosols play a role in the elevated aerosol concentrations in the polar vortex during the northern hemisphere winter of 2020.

We utilised the CALIOP aerosol extinction coefficient night-time retrievals to examine the evolution of the vertical profile of

the plume. We initially observed two distinct peaks just above and just below the average tropopause height before a singular

peak at around 14km was observed from August onwards. The SO2+ash model represents the altitude of the aerosol well throughout the months following the eruption, however the SO2only simulation displayed values of aerosol extinction consistently lower in altitude (Figs. 9 and 10) resulting in a much faster decay rate due to transfer to the troposphere through tropospheric folds. The modelled aerosol extinction peak was prematurely early and overestimated in both scenarios which could be due to the rate of coagulation in the stratosphere or how the model represents new particle formation. After more moderate volcanic eruptions the rate of coagulation can be slower resulting in smaller less optically active particles and a delayed aerosol extinction peak. We see in both model simulations the accumulation mode dominates the column aerosol burden. This could suggest that the rate of transfer from optically inactive Aitken to optically active accumulation mode in the model may be too fast.

Finally the impact of the modelled Raikoke eruption was compared to the HadGEM2 simulation of the Sarychev Peak eruption. The Northern Hemisphere peak mean perturbation to the sAOD after the Sarychev Peak eruption was 0.01 compared to 0.03 after the Raikoke eruption. The greatest radiative forcing on the Northern Hemisphere after the Raikoke eruption was 0.48 Wm$^{-2}$, almost double that seen in the July following the Sarychev Peak eruption, 0.24 Wm$^{-2}$, despite only a 25% increase in the injected $SO_2$ amount. Whilst the eruptions were similar in altitude and latitude, the impact seen in UKESM1 is far greater in Raikoke than after Sarychev Peak. Some of these differences may stem from the large dependence on meteorology that has been noted for low-altitude eruptions (Jones et al., 2016), with differences in how the $SO_2$ was injected into the model contributing to this. We also compare an SO2only Raikoke-only simulation and find a Northern Hemispheric peak AOD of approximately 0.02 but a greater peak radiative forcing of 0.55 Wm$^{-2}$. This is to be expected since the injected ash would contribute a positive radiative forcing counterbalancing the negative radiative forcing from the sulfate aerosol. Finally, UKESM1 represents a major advance from HadGEM2 with a new core physical model including a well-resolved stratosphere and enhanced tropospheric-stratospheric chemistry.

Accurately modelling the evolution of volcanic plumes and therefore identifying the impacts they have on the climate is a difficult task. Our analysis provides several lines of evidence to suggest that including ash in model emission schemes can improve the representation of volcanic plumes in global climate models. Whilst the model is not perfect at representing each process it provides reasonable *e*-folding times for the conversion of $SO_2$ to sulfate aerosol and models the geographic distribution of the aerosol well. Future work might consider internal mixture of sulfate and ash aerosol which could yield differences in the aerosol microphysical and optical properties. Including smoke aerosol in simulations of this eruption would also be useful to identify if this would improve the model performance. A better representation of volcanic ash could also be applied, since this study used mineral dust as a proxy since the refractive indices and size distributions are similar. Ultimately, this study has shown that volcanic emissions are far more complicated than simple injections of $SO_2$ and that limitations in remote sensing observations hamper definitive attribution of particle composition. These results suggest the strong need for in situ sampling of aerosol from instrumentation on airborne observational platforms. While such measurements of ash-sulfate

mixtures have been performed following eruptions that predominantly loaded the troposphere (e.g. Johnson et al., 2021; Turnbull et al., 2012; Newman et al., 2012), the dearth of such measurements in the stratosphere means that definitive attribution of aerosol composition, and microphysical and optical properties remains extremely challenging.

Final caveats to our modelling study include the fact that we do not include any emission of water vapor (e.g. Joshi and Jones, 820    2009) nor of any halogens (e.g. Staunton-Sykes et al., 2020); these species are commonly co-emitted with volcanic eruptions and may influence the oxidation rates of sulfur dioxide and the detailed evolution of the resultant aerosol plume.

## Code and data availability

The data is available at https://doi.org/10.5281/zenodo.7602563 and the code to reproduce the figures is at https://zenodo.org/badge/latestdoi/596982123

## Author Contributions

AW, MO and LDP analysed the satellite and in-situ data and created the combined observational dataset. AJ and JH devised
the experimental set up for UKESM1. AW, AJ and JH analysed the results of the simulations. AW prepared the manuscript with contributions from all co-authors.

## Competing interests

The authors declare that they have no conflict of interest.

**Acknowledgements**

AW was funded via a UKRI Centre for Doctoral Training in Environmental Intelligence PhD studentship hosted by the University of Exeter. JH, AJ, MO was supported by the Met Office Hadley Centre Climate Programme funded by BEIS. JH would also like to acknowledge support from the NERC funded EXTEND project (NE/W003880/1) and from SilverLining through its Safe Climate Research Initiative. LDP was supported from the NERC funded SASSO standard grant (NE/
S00212X/1). JH and DP would like to acknowledge the support of the NERC funded ADVANCE project (NE/T006897/1).

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
