# Peer review of "Including ash in UKESM1 model simulations of the Raikoke volcanic eruption reveal improved agreement with observations"

_EGUsphere, 2022_

## Referee Comment (RC3)

The paper presents an analysis of the UKESM1 model simulations of the Raikoke eruption and compares it to satellite observations. The authors argue that including ash in the model emission scheme, which is usually neglected, provides a more accurate simulation of the evolution of the volcanic plume and a better agreement with observations.

The paper is well written and contributes significantly to modelling the impact of eruptions on the stratosphere and climate.

I recommend it for publication, subject to some improvement. My main concern is the lack of a proper description of the methods to construct the observation dataset used for comparison, especially CALIPSO. I also find the Radiative impact analysis incomplete and suggest the authors expand the analysis to other months when the aerosol loading peaks.

**Specific comments:**

L49: "(e.g. Cai et al., 2022)". The reference is missing.

L116:" This study uses quality-assured (QA) daily averaged vertical profiles of aerosol extinction (km-1) at 532nm from the Version CALIOP Level 2 data product."
The authors' description of the CALIPSO data used in the study is lacking. The paper also lacks a data availability section, making it challenging to identify the version used. Based on the description given above, one can guess that the authors are using "CAL_LID_L2_05kmAPro-Standard-V4-20", which is then averaged and gridded into a daily file. If that is the case, I find using CALIPSO's L2 data to derive the sAOD zonal mean time series flawed. L2 files only report aerosol extinction for detected layers and fill values for the rest of the profile. Averaging these profiles will bias the mean sAOD toward large values during the first few months when the aerosol signal is strong and toward very low values later, which appears to be the case in Figure 4. Kar et al. (2019) reported that to retrieve profiles of stratospheric aerosol extinction and backscatter coefficients reliably, they relied on substantial spatial and temporal averaging because of CALIPSO's poor signal-to-noise ratio in the stratosphere. Applied vertical averaging was 900 m, 5$^o$ latitude, and 20$^\circ$ longitude for a whole month. They averaged Level 1b backscattered profiles, not level 2 extinction profiles. Others used similar averaging to retrieve their stratospheric aerosol profiles (Thomason et al., 2007; Vernier et al., 2009; Vernier et al., 2011). The authors need to describe better the CALIPSO data and methods used to derive the sAOD. If indeed they used L2 data, they need to acknowledge its limitation when comparing it to OMPS and the model's sAOD.

L119: "Campbell et al. (2012)". The reference is missing.

L152: "80km", change to 40.5km.

L164: "The retrieval issues described here have a significant impact on our study since it was estimated that the initial plume reached altitudes of between 11 and 20km (Vaughan et al., 2021; Osborne et al., 2021)." This sentence contradicts the previous one, which states that the loss of sensitivity below 17 km is only for the short wavelength and SH. Please explain why this is an issue in the NH where this eruption took place.

L293: "The notable difference between the observations and the model is unexpected given that the magnitude of SO2 injected was based on observations (Muser et al., 2020; Kloss et al., 2021; De Leeuw et al., 2021)." Have you considered that the modeled $SO_2$ might compare better with other instruments or datasets, given that cited $SO_2$ volcanic mass was derived using different instruments?

Sections 4.2 and 4.3, Figures 4 and 5s: If the authors used L2 CALIPSO data (see my previous comment), they need to revise this section. OMPS LP and the model provide complete aerosol profiles, unlike CALIPSO L2, which only provides the elevated layers. I also suggest they compare their sAOD to CALIPSO's official L3 data for reference.

Section 4.6 and Figure 9 and L520: "After this, the observations show much of the aerosol plume below the average tropopause height, and by December, the magnitude of aerosol extinction coefficient is negligible."
Simplifying the volcanic aerosol distribution and tropopause altitudes in NH into one profile/one altitude per month can lead to the wrong conclusions. The volcanic plume is transported poleward to lower altitudes but still in the stratosphere, which is not apparent in the figure. The tropopause altitudes can vary significantly between 30-90N (~7 and 15km). Therefore, the conclusion that the plume is moving to the troposphere is inaccurate. The authors should modify the figure by removing the tropospheric part of each profile before calculating the monthly mean profile, thus ensuring that it only represents the stratospheric profile. Similarly, they can also calculate the tropospheric profile if they believe that a significant part of the plume was transported to the troposphere, although I doubt it. They should also modify the text accordingly.

L533: "This could be due to aerosol microphysical processes such as the rate of coagulation and/or condensation or how the model represents new particle formation." I don't understand why the authors are speculating on particle growth. They should be able to investigate the particle size in the model and provide a definitive explanation.

L549: "….Due to more removal processes in the troposphere this results in a much faster decay rate and shorter lifetime." See my previous comment. Please revise this paragraph and any discussion regarding the average tropopause altitude.

Section 4.7 Radiative impact: This section is a significant result of this work; however, the way it is presented is lacking. The RI calculation is only shown for July when Raikoke's sAOD peaked in the later months. The authors need to expand their analysis and table 2 to provide the RI for other months. The analysis did not show when Sarychev's sAOD peaked. Was it in July or later?

L580: "We also note that we cannot make exact global comparisons between our analysis and the Sarychev Peak eruption due to the impacts of the first Ulawun eruption (26th June) in UKESM1" I don't understand why this is not possible. The authors should be able to run the

model simulation without Ulawun and derive the Raikoke-only RI. Please revise the text accordingly.

L584: "The cooling effect is greatest over North America, similar to the distribution of sAOD. This is in contrast to the Sarychev Peak eruption, where the modelled radiative impact indicated a stronger cooling over Russia in comparison to North America." Can you please comment on the cause of this difference?

Section 5. "Discussion and conclusion": the section should be renamed "Summary and conclusion" as it mainly summarizes the analysis and results presented in the previous sections.

L642:" lower in altitude resulting in a much faster decay rate due to transfer to the troposphere through tropospheric folds." See my previous comment regarding the average tropopause and revise the text accordingly.

L655: "Some of these differences may stem from the large dependence on meteorology that has been noted for low-altitude eruptions (Jones et al., 2016)." I don't understand this explanation. The main difference between the two eruptions is the presence of a large amount of ash in Raikoke, not the meteorology. The authors should investigate their model for the exact differences in the aerosol optical properties that result in such differences and provide a better explanation. They could also compare the $SO_2$only RI to the $SO_2$+ash and Sarychev, which is mostly $SO_2$.

L663: "Future work might consider …" The authors should include the addition of smoke aerosol to future work needed to improve their results.

The paper is missing the code and data availability section.

Kar, J., Lee, K.-P., Vaughan, M. A., Tackett, J. L., Trepte, C. R., Winker, D. M., Lucker, P. L., and Getzewich, B. J.: CALIPSO level 3 stratospheric aerosol profile product: version 1.00 algorithm description and initial assessment, Atmos. Meas. Tech., 12, 6173–6191, https://doi.org/10.5194/amt-12-6173-2019, 2019.
Thomason, L. W., Pitts, M. C., and Winker, D. M.: CALIPSO observations of stratospheric aerosols: a preliminary assessment, Atmos. Chem. Phys., 7, 5283–5290, https://doi.org/10.5194/acp-7-5283-2007, 2007.
Vernier, J. P., et al. (2009), Tropical stratospheric aerosol layer from CALIPSO lidar observations, J. Geophys. Res., 114, D00H10, doi:10.1029/2009JD011946.
Vernier, J.-P., et al. (2011), Major influence of tropical volcanic eruptions on the stratospheric aerosol layer during the last decade, Geophys. Res. Lett., 38, L12807, doi:10.1029/2011GL047563.

---

## Community Comment (CC1)

The manuscript by Wells et al. shows model simulations of the Raikoke volcanic eruption in 2019 and comparisons with space-borne observations. They found that volcanic sulfate particles alone are not sufficient in order to explain the high AOT values that were observed during summer, autumn and winter 2019-2020. Including ash into their UKESM1 simulations, however, enhances the agreement between model results and observations. The manuscript is an important contribution to the literature as it shows comparisons of Raikoke modelling results including ash in the simulations vs. observations. There are, however, some questions and comments to parts of the manuscript. Some of the concerns are listed as follows:

Question 1: Why should the Raikoke event regarding the produced stratospheric aerosol be so different compared to Pinatubo, Sarychev and many others.

After these eruptions, the ash was removed quite quickly (within a few weeks) and the sulfate was then left as the only volcanic aerosol type? And now, the ash was present even after months (September –November 2019)?

Question 2: If volcanic ash would be present in the northern hemispheric stratosphere in summer and autumn 2019 (authors write 0.4 – 1.8 Tg) in a comparable amount as sulfate (authors write 1.5Tg ± 0.2 Tg), one should find a lot of cases with enhanced particle depolarization ratios of the stratospheric aerosol layers, for example with CALIOP measurements. To our knowledge this was not the case.

Here are some examples that show low depolarization ratios only:

https://www-calipso.larc.nasa.gov/products/lidar/browse_images/show_v4_detail.php?s=production&v=V4-10&browse_date=2019-07-18&orbit_time=06-56-08&page=4&granule_name=CAL_LID_L1-Standard-V4-10.2019-07-18T06-56-08ZD.hdf

https://www-calipso.larc.nasa.gov/products/lidar/browse_images/show_v4_detail.php?s=production&v=V4-10&browse_date=2019-07-20&orbit_time=04-54-59&page=4&granule_name=CAL_LID_L1-Standard-V4-10.2019-07-20T04-54-59ZD.hdf

Question 3: Why is the potential impact of the record-breaking Siberian fires on the UTLS aerosol load totally ignored (for the period from August 2019 to December 2019)? The smoke certainly influenced the aerosol in the 8-15 km height range for latitudes from 65-

90°N. The Siberian smoke is discussed by Ohneiser et al. (ACP, 2021) and by Ansmann et al. (Frontiers, 2021).

We hypothesize that simulations with sulfate and smoke (instead of sulfate and ash) may even explain better all the OMPS-LP and CALIOP observations.

Question 4: How is the AOD computed in the case of the CALIOP observations. Is it computed from the backscatter profile multiplied by a sulfate lidar ratio (of 40-50sr)? …and then integrated from the tropopause to 20 km height?

Since we believe that the smoke fraction was more than 80-90% (and the sulfate fraction 10-20%, and the ash fraction 0%) at high northern latitudes in the UTLS height range then the appropriate lidar ratio is 80-90 sr according to Ohneiser et al. (2021) and then the CALIOP AOD would be even a factor of 2 larger then shown since September 2019.

Question 5: Are you sure that all the spaceborne passive remote sensing techniques work properly during the maximum of the stratospheric perturbation in August 2019 and September 2019? Maybe the channels were almost, partly or totally saturated or almost saturated, at least strongly biased?

Table 1 is not useful, AERONET data are biased by the tropospheric impact, measurements at 19-20°N were certainly influenced by the Ulawun eruption. It is at least impossible to state the Ulawun eruption had no effect. What about an MLO lidar? If there is a lidar, what depolarization ratio was measured, what about the observed AOD?

Is the Brewer-Dobson Circulation already so strong in August and September to explain such a strong transport of aerosol towards the North Pole?

Question 6: According to Fig. 9, the initial aerosol optical thickness in summer would be overestimated when assuming a sulfate + ash mixture. However, already from October onwards the high observed AOT values cannot be explained with sulfate + ash only; and of course, not at all with sulfate only. The decay times of sulfate and sulfate + ash seem to be way to short in order to describe the high AOTs in late 2019 and early 2020. The decay time of wildfire smoke is longer and the authors should mention that especially the Arctic aerosol situation was more influenced by wildfire smoke particles. A significant amount, especially north of 65-70°N the dominating amount of the stratospheric aerosol type must have been Siberian wildfire smoke particles, as Ohneiser et al. 2021, ACP and Ohneiser et al. 2022,

ACPD write.

Figure 9 is confusing. A mean tropopause for the latitudes from 30-90°N makes no sense…. A UTLS layer, one half in the upper troposphere and one half in the stratosphere, is not what is expected after a moderate eruption such as the Raikoke eruption. One expects that the layer is fully and clearly above the tropopause. And all the aerosol in the upper troposphere (at latitudes below about 55°N) should have been be efficiently removed by cirrus clouds. In addition, we speculate, nobody knows the exact SO2 injection heights? Thus, all the simulations seem to be just playing around with possibilities.

---

## Author Comment (AC2)

We would like to thank Kevin Ohneiser for taking the time to read and provide comments on our paper. We believe that changes made to the paper based on his and the reviewers' comments have improved the manuscript.

We have addressed his comments (in black) below in red.

**Response to Kevin Ohneiser**

The manuscript by Wells et al. shows model simulations of the Raikoke volcanic eruption in 2019 and comparisons with space-borne observations. They found that volcanic sulfate particles alone are not sufficient in order to explain the high AOT values that were observed during summer, autumn and winter 2019-2020. Including ash into their UKESM1 simulations, however, enhances the agreement between model results and observations. The manuscript is an important contribution to the literature as it shows comparisons of Raikoke modelling results including ash in the simulations vs. observations. There are, however, some questions and comments to parts of the manuscript. Some of the concerns are listed as follows:

Question 1: Why should the Raikoke event regarding the produced stratospheric aerosol be so different compared to Pinatubo, Sarychev and many others.

After these eruptions, the ash was removed quite quickly (within a few weeks) and the sulfate was then left as the only volcanic aerosol type? And now, the ash was present even after months (September –November 2019)?

Thank you for your question. The role of ash in the dispersion of volcanic aerosol after an eruption has not only been studied for the Raikoke eruption but also for Pinatubo (e.g. Shallcross et al., 2021; Stenchikov et al., 2021; Abdelkader et al., 2023). Ground-based and airborne lidar observations (Browell et al., 1993; Vaughan et al., 1994) suggested the base of the Mount Pinatubo volcanic plume contained ash particles coated in sulphuric acid for approximately 9 months after the eruption. More recently, after the Mt. Kelud eruption, observations suggested that ash-rich aerosol accounted for part of the aerosol plume 3 months following the eruption (Vernier et al., 2016).

It is also quite possible that the Sarychev Peak eruption also contained a significant amount of volcanic ash. If you consider the stratospheric AOD shown in Figure 5 of Haywood et al. (2010), it becomes immediately clear that the observations show a far longer e-folding time than the model. It is quite possible that including ash in simulations of Sarychev Peak would have led to an improved agreement with observations from the OSIRIS limb-sounding instrument. Definitively attribution of any improvement would be difficult because the simulations of Haywood et al. (2010) were performed with an older version of the model

(HadGEM2-ES) which utilised a different aerosol scheme and many other different processes and parameterisations.

[Figure]

**Figure 5.** Time versus latitude plots of the aerosol optical depth at 750 nm from (a) HadGEM2 and (b) OSIRIS. The pre-eruption zonal mean AOD is removed from the OSIRIS data to give the perturbation to the AOD.

Question 2: If volcanic ash would be present in the northern hemispheric stratosphere in summer and autumn 2019 (authors write 0.4 – 1.8 Tg) in a comparable amount as sulfate (authors write 1.5Tg ± 0.2 Tg), one should find a lot of cases with enhanced particle depolarization ratios of the stratospheric aerosol layers, for example with CALIOP measurements. To our knowledge this was not the case.

Here are some examples that show low depolarization ratios only:

https://www-calipso.larc.nasa.gov/products/lidar/browse_images/show_v4_detail.php?s=production&v=V4-10&browse_date=2019-07-18&orbit_time=06-56-08&page=4&granule_name=CAL_LID_L1-Standard-V4-10.2019-07-18T06-56-08ZD.hdf

https://www-calipso.larc.nasa.gov/products/lidar/browse_images/show_v4_detail.php?s=production&v=V4-10&browse_date=2019-07-20&orbit_time=04-54-59&page=4&granule_name=CAL_LID_L1-Standard-V4-10.2019-07-20T04-54-59ZD.hdf

With the greatest of respect, we present considerable evidence that the size distribution is perturbed which can only be through the presence of volcanic ash. We utilise the OMPS-LP Angstrom exponent for the first 30 – 40 days (see Figure 8 of the manuscript).

[Figure]

**Figure 8:** Daily evolution of Ångström Exponent for OMPS-LP (blue), UKESM1 SO2only (red) and UKESM1 SO2+ash (dark red). Calculated using area weighted sAOD between 30–90° N (same as Fig. 7) using 510nm and 869nm wavelengths.

We note that the observations do not suggest an extended influence in 'Autumn', i.e. September, October, November. The e-folding time for the Angstrom exponent suggest that the depolarising ash has been removed from the atmosphere within the first 40 days, while the model suggest that it is removed over a shorter period (about 20days). Note that the values of Angstrom exponent from around days 50 – 300 may be smaller than that after day 300 because sulfate aerosol under volcanic eruptions will exhibit larger sizes (and hence smaller Angstrom exponent) than background sulfate because of enhanced coagulation. This has been known from measurements during quiescent and eruption influenced measurements from balloon-borne optical particle counters for several decades (e.g. SPARC report; SPARC Report No.4 | SPARC (sparc-climate.org), their Figure 1.2, where the accumulation mode gradually decreases in number relative to the Aitken mode).

The OMPS-LP data suggest that super-micron ash is present in the stratosphere for around 40 days. We present below examples of CALIOP stratospheric aerosol layers with enhanced depolarization ratios. The first is from 30th June, nearly 10 days after the eruption.

[Figure]

[Figure]

[Figure]

This aerosol has come from the volcanic eruption (Figure 1) due to the timing and location that we have chosen. It has certainly not come from the Siberian wildfires as they didn't begin until 19th July. CALIOP identifies this mostly as dust but also includes other aerosol subtypes including polluted dust, elevated smoke and volcanic ash.

Almost 20 days after the eruption we present another CALIOP swath across the north pacific. We again see a combination of aerosol subtypes identified here including dust, polluted dust, elevated smoke and sulfate.

[Figure]

[Figure]

We further present a final CALIOP swath almost 30 days after the eruption. Even after 30 days there are still a combination of aerosol subtypes identified by CALIOP including dust, elevated smoke, volcanic ash and sulfate.

[Figure]

We conclude that, as evidenced from the OMPS-LP, super-micron dust/ash is present in significant quantities for the first 30 – 40 days after the eruption. In each of the example's dust (yellow) was identified by CALIOP due to larger depolarizing ratios.

Question 3: Why is the potential impact of the record-breaking Siberian fires on the UTLS aerosol load totally ignored (for the period from August 2019 to December 2019)? The smoke certainly influenced the aerosol in the 8-15 km height range for latitudes from 65-90°N. The Siberian smoke is discussed by Ohneiser et al. (ACP, 2021) and by Ansmann et al. (Frontiers, 2021).

We hypothesize that simulations with sulfate and smoke (instead of sulfate and ash) may even explain better all the OMPS-LP and CALIOP observations.

Thank you for highlighting this. The manuscript has been updated to include information about the Siberian and Canadian forest fires. As you say, this is a hypothesis, but far from proven at present. For example, the study of Boone et al (2022) conclude, "Contrary to previous reports, the aerosol blanket was not comprised of smoke particles". A sensitivity study including sulfate and smoke would be interesting but something for future work.

Question 4: How is the AOD computed in the case of the CALIOP observations. Is it computed from the backscatter profile multiplied by a sulfate lidar ratio (of 40-50sr)? ...and then integrated from the tropopause to 20 km height?

Since we believe that the smoke fraction was more than 80-90% (and the sulfate fraction 10- 20%, and the ash fraction 0%) at high northern latitudes in the UTLS height range then the appropriate lidar ratio is 80-90 sr according to Ohneiser et al. (2021) and then the CALIOP AOD would be even a factor of 2 larger then shown since September 2019.

The AOD is calculated using the CALIPSO L2 aerosol extinction coefficient at 532nm. We use the observed tropopause height and integrate above this to calculate the stratospheric aerosol optical depth. We have acknowledged the limitations of using this data in the updated manuscript.

Question 5: Are you sure that all the spaceborne passive remote sensing techniques work properly during the maximum of the stratospheric perturbation in August 2019 and September 2019? Maybe the channels were almost, partly or totally saturated or almost saturated, at least strongly biased?

We acknowledge in the text that there are limitations to using a limb-profiler for the months immediately following the eruption. This is taken into account when we create the combined observational dataset in section 4.3.

Table 1 is not useful, AERONET data are biased by the tropospheric impact, measurements at 19-20°N were certainly influenced by the Ulawun eruption. It is at least impossible to state the Ulawun eruption had no effect. What about an MLO lidar? If there is a lidar, what depolarization ratio was measured, what about the observed AOD?

Since the AOD is elevated for three months at 95% confidence we disagree that Table 1 (now Table 3) is not useful. The likelihood of this happening by change (assuming that each month is independent) would be approximately $0.05^3$. We acknowledge the impact of the Ulawun eruption on the Mauna Loa observations in the text. We also compared the two UKESM1 model simulations (SO2only and SO2+ash) to those performed as Raikoke-only (removing the Ulawun injections) and noted a negligible influence ($<0.3 \times 10^{-3}$) from the eruption in the MLO region.

Is the Brewer-Dobson Circulation already so strong in August and September to explain such a strong transport of aerosol towards the North Pole?

The transport of the plume and therefore the impact on the climate can be dependent on a multitude of parameters, including the location of the volcano and the local meteorological variability (e.g. Jones et al., 2017). We can also see from the observations of sulfate aerosol in the third CALIOP swath which we provide in this document, that the plume has travelled northwards of 60N by 18th July.

Question 6: According to Fig. 9, the initial aerosol optical thickness in summer would be overestimated when assuming a sulfate + ash mixture. However, already from October onwards the high observed AOT values cannot be explained with sulfate + ash only; and of course, not at all with sulfate only. The decay times of sulfate and sulfate + ash seem to be way to short in order to describe the high AOTs in late 2019 and early 2020. The decay time of wildfire smoke is longer and the authors should mention that especially the Arctic aerosol situation was more influenced by wildfire smoke particles. A significant amount, especially north of 65-70°N the dominating amount of the stratospheric aerosol type must have been Siberian wildfire smoke particles, as Ohneiser et al. 2021, ACP and Ohneiser et al. 2022,ACPD write.

Thank you for highlighting this. Discussion around the influence of the Siberian and Canadian wildfires has been included in the revised manuscript. However, as there is conflicting scientific evidence e.g. the analysis by Boone et al. (2022) of ACE data, we address both sides of the argument surrounding the impact of the Siberian and Canadian wildfires.

Figure 9 is confusing. A mean tropopause for the latitudes from 30-90°N makes no sense.... A UTLS layer, one half in the upper troposphere and one half in the stratosphere, is not what is expected after a moderate eruption such as the Raikoke eruption. One expects that the layer is fully and clearly above the tropopause. And all the aerosol in the upper troposphere (at latitudes below about 55°N) should have been be efficiently removed by cirrus clouds. In addition, we speculate, nobody knows the exact SO2 injection heights? Thus, all the simulations seem to be just playing around with possibilities.

We agree that it is difficult to identify the transport of the aerosol under an area average. Therefore we have included a new figure which shows the monthly aerosol extinction coefficient as a function of latitude and altitude. This gets round the problem of showing a single line for the height of the tropopause (albeit with the variability also indicated). In this figure we do see some transfer of aerosol from the stratosphere to the troposphere, mostly at the midlatitudes. This section has been rewritten to include the new analysis.

References:

Abdelkader, M., Stenchikov, G., Pozzer, A., Tost, H., and Lelieveld, J.: The effect of ash, water vapor, and heterogeneous chemistry on the evolution of a Pinatubo-size volcanic cloud, Atmos. Chem. Phys., 23, 471-500, 10.5194/acp-23-471-2023, 2023.

Boone, C. D., Bernath, P. F., Labelle, K., and Crouse, J.: Stratospheric Aerosol Composition Observed by the Atmospheric Chemistry Experiment Following the 2019 Raikoke Eruption, Journal of Geophysical Research: Atmospheres, 127, e2022JD036600, https://doi.org/10.1029/2022JD036600, 2022.

Browell, Edward V., Carolyn F. Butler, Marta A. Fenn, William B. Grant, Syed Ismail, Mark R. Schoeberl, Owen B. Toon, Max Loewenstein, and James R. Podolske. "Ozone and aerosol changes during the 1991-1992 Airborne Arctic Stratospheric Expedition." Science 261, no. 5125 (1993): 1155-1158.

Haywood, J. M., Jones, A., Clarisse, L., Bourassa, A., Barnes, J., Telford, P., Bellouin, N., Boucher, O., Agnew, P., Clerbaux, C., Coheur, P., Degenstein, D., and Braesicke, P.: Observations of the eruption of the Sarychev volcano and simulations using the HadGEM2 climate model, Journal of Geophysical Research, 115, 10.1029/2010jd014447, 2010.

Jones, A. C., Haywood, J. M., Dunstone, N., Emanuel, K., Hawcroft, M. K., Hodges, K. I., and Jones, A.: Impacts of hemispheric solar geoengineering on tropical cyclone frequency, Nat Commun, 8, 1382, 10.1038/s41467-017-01606-0, 2017.

Shallcross, S., Mann, G., Schmidt, A., Haywood, J., Beckett, F., Jones, A., Neely, R., Vaughan, G., and Dhomse, S.: Long-lived ultra-fine ash particles within the Pinatubo volcanic aerosol cloud and their potential impact on its global dispersion and radiative forcings, April 01, 2021, 10.5194/egusphere-egu21-16034, 2021.

Stenchikov, G., Ukhov, A., Osipov, S., Ahmadov, R., Grell, G., Cady-Pereira, K., Mlawer, E., and Iacono, M.: How Does a Pinatubo-Size Volcanic Cloud Reach the Middle Stratosphere?, Journal of Geophysical Research: Atmospheres, 126, e2020JD033829, https://doi.org/10.1029/2020JD033829, 2021.

Vaughan, G., D. P. Wareing, S. B. Jones, L. Thomas, and N. Larsen. "Lidar measurements of Mt. Pinatubo aerosols at Aberystwyth from August 1991 through March 1992." Geophysical research letters 21, no. 13 (1994): 1315-1318.

Vernier, Jean-Paul, T. Duncan Fairlie, Terry Deshler, Murali Natarajan, Travis Knepp, Katie Foster, Frank G. Wienhold, Kristopher M. Bedka, Larry Thomason, and Charles Trepte. "In situ and space-based observations of the Kelud volcanic plume: The persistence of ash in the lower stratosphere." Journal of Geophysical Research: Atmospheres 121, no. 18 (2016): 11-104.

---

## Author Response (AR2)

**Response to reviews of "Including ash in UKESM1 model simulations of the Raikoke volcanic eruption reveal improved agreement with observations" by Wells et al.**

We extend our thanks to the three reviewers for their thorough evaluation of our work and the careful reviews and help in improving this paper. We are grateful to the reviewers for their time in reviewing the paper and their detailed responses. We would also like to thank Kevin Ohneiser for his comments, we believe that the changes made to the paper will satisfy his questions as many were similar to those in the three reviewers comments.

We are glad that reviewers liked the paper and consider it worth publishing after addressing their points.

We have addressed the reviewers' comments (in black),  as per our responses (in red).

**Response to reviewer 1.**

The paper analyzes observations and models the Raikoke eruption in June 2019 in the Kuril Islands. The authors use the state-of-the-art general circulation model with the well- developed aerosol and stratospheric chemistry module in combination with the best available observations. They argue that the existing observations are still insufficient to describe the essential elements of the volcanic cloud evolution reliably. The model simulations help to better understand the overall evolution of the volcanic cloud, e.g., the simulated optical depth is twice higher than the observed one, but simulated SO2 loadings are significantly lower than that observed by OMPS-NM. The results are presented in a somewhat qualitative way. More quantitative information would help.

The paper is generally well-written and logically organized. However, it requires a more rigorous description of physical processes and how they are parameterized in the model.

The literature review is complete. I suggest the authors look at two recently published papers that consider the ash radiative heating and lofting effect in detail and ash coating by sulfate and heterogeneous chemistry.

Stenchikov, G., Ukhov, A., Osipov, S., Ahmadov, R., Grell, G., Cady Pereira, K., ... & Iacono, M. (2021). How Does a Pinatubo Size Volcanic Cloud Reach the Middle Stratosphere? Journal of Geophysical Research: Atmospheres, 126(10), e2020JD033829.

Abdelkader, M., Stenchikov, G., Pozzer, A., Tost, H., & Lelieveld, J. (2022). The effect of ash, water vapor, and heterogeneous chemistry on the evolution of a Pinatubo-size volcanic cloud. Atmospheric Chemistry and Physics Discussions, 1-49.

Thank you for these paper suggestions. We have found them very insightful and have included them in the revised manuscript

L100: Please formulate the objectives and science questions you are targeting in this study

Corrected – we now provide a brief description of the scientific objectives.

L147-160: CALIOP should be discussed separately in line with ONPS-NM and OMPS_LP

Corrected – we have now revised the section describing the CALIOP data processing.

Section 3.1: You have to provide a more detailed description of the model physics, especially microphysical parameterizations.

Thank you for highlighting this missing area. A more detailed description has been added to Section 3.1. This includes a description of the UKCA aerosol scheme and a brief description of the mineral dust scheme. The mineral dust scheme is modified to provide a suitable proxy for volcanic ash as detailed in section 3.2. We also now provide brief reference to other simulations of stratospheric sulfate aerosol plumes which show that the simulations are broadly consistent with other models.

Section 3.2: Please discuss the size distribution of emitted ash explicitly. What was the refractive index of ash? The deposition of dust particles should be corrected to be applicable to the stratospheric conditions.

The size distribution and refractive index have been added to the text. The parameterization for gravitational settling depends upon the gravitational settling velocity, and the aerodynamic (and surface) resistances (https://www.ukca.ac.uk/images/e/eb/Umdp84_vn82.pdf) and, in principle, is applicable to both the troposphere and the stratosphere. We do not make any modifications or adjustments to this parameterisation as the objectives are to assess the performance of the base-state of the model.

Section 3.3 is out of context. It is better to reference Mie calculations in the place where they are used.

We have dropped the heading and redefined the previous section heading to be: "Simulations and reference wavelengths for model/observation intercomparisons". This provides the reader with enough information on how re-scaling from different wavelengths is performed, without cluttering the results with a methodological description within the results section

L222: "logical timeline" ?

Re-worded

Figure 1: The simulated SO2 loadings are more dispersed than in observations. Could you please comment on this? Could you provide statistical scores for the analysis?

The text has been updated to include a comment on this. A more quantitative analysis of this is shown in Fig 2 and Table 2.

Figure 4: Please label the panels in figure 4.

Apologies – corrected.

L391: Statistically significant at what confidence level?

Corrected – 5% level

L439: "Figure 7 begins to illustrate" - could you say it better?

Reworded.

L450-459: These are pretty vague explanations. Could you please clarify?

Wording changed to clarify

Figure 7: The e-folding time for SO2-only simulations is more consistent with observations than SO2+ash simulations, which decay faster than observed. Could you please comment on this?

The e-folding times for the SO2only simulations with and without observational limits imposed are 45 and 52 days respectively. However, the e-folding times for the SO2+ash simulations with and without observational limits imposed are 90 and 101 days respectively. The e-folding time for the combined observational data set is 145 days, longer than both model simulations, however SO2only simulations decay faster than SO2+ash and observed, therefore SO2+ash is the more consistent model to the combined observational data set. The wording within section 4.4 has been updated for clarity.

L556-558: What additional analysis?

"Additional" removed

Section 4.7: Do you talk about SW clear sky radiative forcing at the top of the atmosphere? Please make it clear. Is the LW forcing negligible?

Apologies, this was not made clear. The text has been amended for more clarity as below.

Haywood et al., 2010 ran simulations of the Sarychev eruption over a three-month period June-August 2009 and explored the difference between the aerosol optical depth and radiative forcing due to anthropogenic aerosol and the Sarychev Peak stratospheric aerosol plume. They found that in some regions of the NH the AOD was of comparable magnitude or greater than the AOD from anthropogenic emissions. We estimate the impact of the Raikoke eruption on the approximate radiative forcing exerted by the volcanic aerosol and compare this to the Sarychev Peak eruption. In Figure 11 we have recreated the Sarychev Peak AOD and radiative forcing plots from Haywood et al., 2010 Figure 12 using data from HadGEM2 and compare them against our UKESM1 Raikoke simulations, eliminating the impacts of the Ulawun eruption. As in the results presented by Haywood et al. (2010), our UKESM1 simulations are nudged to reanalysis data, but the evolution of the control and experiment simulations differ slightly and result in some differences in the cloud and fields. Due to this, the all-sky radiative forcing cannot be accurately determined from the difference in the top of the atmosphere shortwave upwelling radiation with and without aerosols. The all-sky shortwave radiative forcing can be approximated for purely scattering aerosol, following the calculation used by Haywood et al., (2010), using the following equation, Eq. (2):

$$Radiative\ forcing \approx (SW_{CTL} - SW_{EXP}) * (1 - A_c), \tag{2}$$

where $SW$ is the clear-sky outgoing shortwave radiation (Wm$^{-2}$) at the top of the atmosphere (TOA) for the control (CTL) and experiment (EXP) and $A_c$ is the cloud fraction (Haywood et al., 1997). This assumes that the contribution from longwave radiation and the radiative forcing from cloudy areas is negligible. It has been shown through radiative transfer calculations that for conservative scattering from sulfate aerosols for optically thick cloud that this is a reasonable assumption (e.g. Haywood and Shine, 1997).

How do you compare Sarychev and Raikoke's forcings? Do you compare them at the month when they are at maximum? Please clarify this.

Apologies, this has been changed so that we compare the forcings at their peak, Sarychev in July and Raikoke in August. We now provide estimates of the radiative forcing evolution in Table format. Unfortunately data from the Sarychev simulations is only available for three months. We have re-written this section quite significantly as there are some interesting findings; it appears that it is not the inclusion of the ash that leads to the stronger radiative forcing, but differences in the base-model formulation, the aerosol scheme and the meteorology. This is now explicitly stated in the revised manuscript.

Section 5: You added ash in simulations and did not show any figure that would characterize ash evolution. How fast does ash deposit? What is its optical depth? There should be almost no ash (if it is not pumice) in the volcanic cloud two weeks after the eruption. Please elaborate on this.

A supplementary figure has been added to discuss the impact of ash evolution further. Text has been modified in section 4.6 and 5.

Also, what about water vapor injections? Are there any estimates of injected water vapor?

There is a lot of current interest in the impacts of water vapour from volcanic eruptions from the recent Hunga Tonga eruption. However, Raikoke was essentially an above water eruption rather than a submarine eruption, and, in common with many other simulations of volcanic plumes, the emission of water vapour was not included in our simulations. We are not aware of any robust estimates of water vapor (nor halogen) emissions. We now caveat these facts at the end of the discussion and include a couple of relevant references.

L602: In the model, $SO_2$ disperses faster than in the real world. What is the conclusion from this?

This is due to a combination of model parameterisations which may lead to excessive diffusion especially in coarse resolution models. While the driving meteorology from ERA5 is at approximately 9km resolution, applying such reanalyses to an N96 coarse resolution model can lead to spurious transport, particularly in areas with strong gradients in horizontal velocity. As noted in the text, additional discrepancies may occur when the extinction from aerosols detection limits falls below detection thresholds. Text changed accordingly

L614-616: It isn't easy to compare with the point observations in MLO, as the cloud is far not uniform at that latitude.

We agree that the cloud is not uniform at that latitude. However, the observations taken from MLO are not point observations due to the viewing geometries of the instrument which makes measurements over the full range of solar zenith angles and hence through considerable slant paths. Additionally, the temporal resolution of our analysis is aggregated into monthly data. There is strong evidence in the observations that the AOD is elevated at beyond 95% statistical significance values for September, October and November. Whilst we understand it is not a perfect comparison, it does provide a useful additional source of data to help understand the contrasting satellite observations

We include some comments on the fact that AERONET data should not necessarily be considered to be point measurements. "Note that AERONET retrievals of sAOD are not a point measurement as they are a function of the solar zenith angle. For solar zenith angles of 60-80 degrees and assuming that any aerosol is located in the lowest 20km

above the observatory, aerosol within a 35-115km radius of the Mauna Loa observatory is included in the observations."

L634-635: The model overestimates the observed sAOD two times. This has to be acknowledged and explained.

The model simulations show the accumulation mode increases quickly after the eruption, dominating the column aerosol burden. We believe that in reality it would take longer for accumulation mode particles to form and that there should be more aerosol in the nucleation mode of the model earlier after the eruption. This is discussed in section 4.6.

L652-655: Why do Sarychev and Raikoke SW clear sky radiative forcings not scale with the amount of injected SO2? Both forcings are obtained in the model, so you can analyze them and tell why it happens. This type of analysis would be helpful.

We now provide a more detailed assessment of the differences between the Raikoke and the Sarychev simulations.

**Response to reviewer 2.**

**General comments**

The paper analyses simulations with a state of the art earth system model including chemistry and satellite observations concerning the effects of a medium size volcanic eruption. It considers 2 satellite instruments providing information on the vertical distribution of aerosol but still focuses too much on the horizontal distribution of vertical integrals in the initial phase where the uncertainty is largest due to lack of observations. Possible interactions with smoke from forest fires are mentioned but not included in the simulations.

The model description is too short. Concerning radiative effects, the standard quantities and notations (e.g. radiative forcing) should be used and not only subsets.

We have added a more comprehensive description of the model

The parts on MLO-AERONET might be skipped because of large uncertainties.

We disagree – since the AOD is elevated for three months at 95% confidence. The chances of this happening by chance (assuming each month is independent) would be approximately $0.05^3$. Thus we contend that the uncertainties are not large and that a robust statistically significant change in the stratospheric AOD is detected. AERONET has become a standard by which global models are judged for tropospheric aerosol modelling, and this approach has been used before for the Sarychev eruption (Haywood et al., 2010). The table from Haywood et al (2010) is reproduced below:-

**Table 1.** AOD at 440, 500, and 675 nm for July and August 2009 and the Mean Obtained From the Period 1996–2008[a]

| Wavelength (nm) | 2009 | Mean 1996–2008 Level 2.0 Data | 2009–Mean |
|---|---|---|---|
| | *July* | | |
| 440 | 0.028 | 0.015 ± 0.006 | +0.013 |
| 500 | 0.022 | 0.011 ± 0.005 | +0.011 |
| *532* | *0.020* | *0.010 ± 0.004* | *+0.010* |
| *550* | *0.019* | *0.010 ± 0.004* | *+0.009* |
| 675 | 0.013 | 0.006 ± 0.004 | +0.007 |
| | *August* | | |
| 440 | 0.024 | 0.013 ± 0.006 | +0.011 |
| 500 | 0.018 | 0.010 ± 0.008 | +0.008 |
| *532* | *0.016* | *0.008 ± 0.004* | *+0.008* |
| *550* | *0.015* | *0.008 ± 0.004* | *+0.007* |
| 675 | 0.009 | 0.005 ± 0.004 | +0.004 |

[a]The values at 532 and 550 nm shown in italics are derived by interpolation from the measured Ångström exponent. The ± figures represent twice the standard deviation (the 95% significance level). The difference between the 2009 values and the means is also shown.

At 550nm, there are perturbations of +0.010 and +0.008 in the AOD for July and August. It appears that any equatorward transport for the Sarychev eruption was quicker than for the Raikoke eruption when peak perturbations of +0.012 were recorded in September. Again, this shows the significant differences in inter-eruption distributions of aerosol. Words are added to this effect.

The quality of the figures should be improved. The selected color scheme is almost as bad as a grey-scale. The paper should be published after revision.

Colour scheme changed.

**Specific comments**

Line 38ff: Here again only estimates for total injections of $SO_2$ are cited. Satellite observations (e.g. Glantz et al., 2014, Höpfner et al., 2015) show that stratospheric aerosol optical depth and $SO_2$ concentration of Sarychev are larger than that of Kasatochi, because Kasatochi has a larger fraction staying in the troposphere (despite of the opposite behavior of the total).

Thank you for your comment. This was useful - we now include the Höpfner et al (2015) reference.

Line 90: Here already 'pumice' should be mentioned, section 5 is too late.

More discussion around pumice has been included, particularly around the assumptions about density.

Line 102: Heating by soot from forest fires might be mentioned here, especially if a sensitivity study is included.

A sensitivity study is outside of the bounds of this study. Text has been adjusted to include suitable references to recent work on lofting caused by black carbon soot within smoke aerosol plumes from intense pyrocumulonimbus events.

Line 173: At Mauna Loa can be effects of local volcanoes. Also a possible signal might be dominated by Ulawun.

The Mauna Loa observatory is much higher than any degassing volcanic activity that was occurring at the time. The eruption from Ulawun was at 5°S, and thus, the majority of the resulting aerosol was trapped in the tropical pipe south of 20N, i.e. south of Mauna Loa (e.g. Kloss et al., 2020). However, we keep this analysis and now show the perturbation from Sarychev derived by Haywood et al (2010). It is noticeable that there was a readily detectable signal for Sarychev as early as July. This strongly suggests that the transport processes are different for the two eruptions, even though many of the input variables (latitude, longitude, timing, altitude, injection amount, vertical profile) are similar. We also compared the two UKESM1 model simulations (SO2only and SO2+ash) to those performed as Raikoke only (removing the Ulawun injections) and noted a negligible influence ($<0.3 \times 10^{-3}$) from the eruption in the MLO region.

Line 188: Here more details on the aerosol module (modal or sectional, microphysics) should be provided. The sentence in the abstract (2 moments for what?) is not enough.

Thank you for highlighting this area. More details on the aerosol module have been included in Section 3.1.

Line 192: Is CNTL without explosive volcanoes only or without any volcano including the ones outgassing into the troposphere?

CNTL is without explosive volcanoes only, long-term outgassing emissions are included. This has been clarified in the text.

Line 194ff: Here is a big source of uncertainties. Are the injections equally distributed in a slab like assumed by Mills et al. (2016)? Or is there a vertical distribution derived from

observations? If yes, mention instrument. How thick is the slab in the upper troposphere? More details please, for $SO_2$ as well as for ash.

The injections are equally distributed between 13-15km plus a single point injection at 10km. This has been made clearer in the text.

Line 214ff: Section 3.3 should be merged with section 3.1, also with respect to the assumed size distribution.

This section has been moved to 3.2.

Line 229f: Please provide a figure for the OMPS-NM background and the background $SO_2$-burden in CNTL in an electronic supplement.

A supplementary figure has been included.

Line 293ff and Figure 3: Don't expect agreement for the initial peak in burden. The e-folding time for OMPS-NM looks smaller than the number in the legend because of the secondary peak. Is OMPS-LP used for the vertical separation? Averaging over half of the northern hemisphere might introduce a lot of artifacts from data gaps (and maybe a mismatch in convection patterns when subtracting results of 2 simulations, depending on the calculation method). What causes the spike at day 60? It might be useful also to show averaged vertically integrated values from OMPS-LP. Please provide more information on the "background" (extra lines here or a figure in a supplement).

We don't use OMPS-LP for the vertical separation. There are no data gaps in the observed longitudinally averaged data so averaging over 30-90 should avoid artifacts from any data gaps. The spike at day 60 is due to removing the long-term background, where a small increase is seen. Extra lines added to text to clarify.

Line 368 and Figure 5: Provide a number or a curve for the longtime background to allow for quantitative comparisons. Unfortunately that is often a problem in the literature.

Long term background sAOD for OMPS-LP: 0.0041

Long term background sAOD for CALIOP: 0.0003

Added to text of the Figure Caption.

Table 1: Is there something wrong with the presentation of the CALIOP-data here? You may remove the column since it is said in the text that CALIOP sees no signal.

Apologies – we agree. CALIOP data removed from table.

Line 421: This is not valid for the latitude of MLO, see Table 1.

This was a contradictory statement. It has been changed for clarity.

Figure 6: Can the bias between the left and right columns be related to background removal?

The figure below is the same as Fig 6 without the background removal and we still see a difference in magnitude between the two satellite datasets.

[Figure]

Line 430ff: This is difficult to understand. How are limits applied? By scaling or truncating of a model quantity, e.g. sAOD? Or by sampling only the regions with data? The jumps in Figure 7 at the time of switching between the instruments look odd. Linear interpolation of what? More details and clarification please.

Apologies for the lack of clarity – we conditionally sample the model data where the observational data is available and apply appropriate detection limits to the model data. The jumps in Figure 7 are due to the differences in observational limits. While the graphical presentation might look better if we simply interpolated over this area, we feel that this would be mis-representing our data analysis process. Instead we have added an annotation that explains that this dip is caused by the model data falling below the CALIOP detection limit threshold. This disappears when we swap to the OMPS-LP data

This has been reworded to clarify as below:

To make consistent and accurate comparisons between the combined satellite dataset and the model simulations it is appropriate to implement the same method used in Fig. 4 and apply minimum detection and spatial limits to the model data. Figure 7 compares the combined OMPS-LP and CALIOP dataset to the two model simulations both with (solid lines) and without (dashed lines) the respective observational limits applied. For the first 4 months after the eruption the combined dataset uses the CALIOP observations, therefore we use the CALIOP minimum detection threshold (0.012 km$^{-1}$) to filter out aerosol extinction values

in the model simulations before calculating the sAOD for comparison. After this we apply the limits of the OMPS-LP data to the model simulations in a similar fashion since OMPS-LP is used in the combined observational dataset for the following months. For both SO2only and SO2+ash we only include model data where observational data were available. When the observational limits were applied, no linear interpolation across the location where the combined dataset switches from CALIOP to OMPS-LP was applied to the model simulations owing to the considerable difference in sAOD across the area of interpolation.

Line 506: This points to the presence of pumice or soot particles.

We provide text and a caveat in the discussion.

Line 525: 'and longitude' or 'half of the northern hemisphere'.

Corrected

Line 564ff: This is something like clearsky radiative forcing. However, by subtracting results of 2 GCM-simulations you have always some kind of cloud feedback. Allsky forcing is smaller but difficult to derive by such an approach. Also infrared is not negligible. Please expand on that and use the proper notation.

Apologies, this section was not made clear. Additions to the text have been made for clarity as below:

Haywood et al., 2010 ran simulations of the Sarychev eruption over a three-month period June-August 2009 and explored the difference between the aerosol optical depth and radiative forcing due to anthropogenic aerosol and the Sarychev Peak stratospheric aerosol plume. They found that in some regions of the NH the AOD was of comparable magnitude or greater than the AOD from anthropogenic emissions. We estimate the impact of the Raikoke eruption on the approximate radiative forcing exerted by the volcanic aerosol and compare this to the Sarychev Peak eruption. In Figure 11 we have recreated the Sarychev Peak AOD and radiative forcing plots from Haywood et al., 2010 Figure 12 using data from HadGEM2 and compare them against our UKESM1 Raikoke simulations, eliminating the impacts of the Ulawun eruption. As in the results presented by Haywood et al. (2010), our UKESM1 simulations are nudged to reanalysis data, but the evolution of the control and experiment simulations differ slightly and result in some differences in the cloud and fields. Due to this, the all-sky radiative forcing cannot be accurately determined from the difference in the top of the atmosphere shortwave upwelling radiation with and without aerosols. The all-sky shortwave radiative forcing can be approximated for purely scattering aerosol, following the calculation used by Haywood et al., (2010), using the following equation, Eq. (2):

$$Radiative\ forcing \approx (SW_{CTL} - SW_{EXP}) * (1 - A_c), \tag{2}$$

where $SW$ is the clear-sky outgoing shortwave radiation (Wm$^{-2}$) at the top of the atmosphere (TOA) for the control (CTL) and experiment (EXP) and $A_c$ is the cloud fraction (Haywood et al., 1997). This assumes that the contribution from longwave radiation and the radiative forcing from cloudy areas is negligible. It has been shown through radiative transfer calculations that for conservative scattering from sulfate aerosols for optically thick cloud that this is a reasonable assumption (e.g. Haywood and Shine, 1997).

Changes have been made regarding proper notation.

For the global results for 2019 the number represents the combined effect of Raikoke and Ulawun. Concerning Sarychev and Raikoke/Ulawun you may compare with the instanteneous allsky forcing of Schallock et al. (2021).

The analysis has now been changed to compare a Raikoke only simulation against the Sarychev eruption.

Line 597: and OMPS-LP.

Corrected

Line 602f: It might not be worth, to repeat this uncertain difference here. Please shorten.

Corrected

Line 630ff: There should be also important contributions from Canadian forest fires as indicated in Osborne et al. (2022). A sensitivity study with volcanic ash and forest fire soot would be of interest, also in the light of the CALIOP observations in the contribution to the discussion by Ohneiser, but maybe in a separate paper if that causes a too long delay. At least mention findings of Osborne et al., (2022) on the effect of soot.

Thank you for highlighting this. The text has been updated to include information about the Canadian forest fires. A sensitivity study would be interesting but something for future work.

Line 647: Caution with these statements. That might be in contradiction to other shown results.

Corrected

**Technical corrections**

Line 49: What 'REFS'? Insert citations.

Corrected

Line 69: Typo in citation.

Corrected

Line 205: Typo.

Corrected

Line 318: zonally averaged or averaged over longitude. Include a-f in Figure.

Corrected

**Response to reviewer 3.**

The paper presents an analysis of the UKESM1 model simulations of the Raikoke eruption and compares it to satellite observations. The authors argue that including ash in the model emission scheme, which is usually neglected, provides a more accurate simulation of the evolution of the volcanic plume and a better agreement with observations.

The paper is well written and contributes significantly to modelling the impact of eruptions on the stratosphere and climate.
I recommend it for publication, subject to some improvement. My main concern is the lack of a proper description of the methods to construct the observation dataset used for comparison, especially CALIPSO. I also find the Radiative impact analysis incomplete and suggest the authors expand the analysis to other months when the aerosol loading peaks.

**Specific comments:**

L49: "(e.g. Cai et al., 2022)". The reference is missing.

Corrected

L116:" This study uses quality-assured (QA) daily averaged vertical profiles of aerosol extinction (km-1) at 532nm from the Version CALIOP Level 2 data product."
The authors' description of the CALIPSO data used in the study is lacking. The paper also lacks a data availability section, making it challenging to identify the version used. Based on the description given above, one can guess that the authors are using "CAL_LID_L2_05kmAPro- Standard-V4-20", which is then averaged and gridded into a daily file. If that is the case, I find using CALIPSO's L2 data to derive the sAOD zonal mean time series flawed. L2 files only report aerosol extinction for detected layers and fill values for the rest of the profile. Averaging these profiles will bias the mean sAOD toward large values during the first few months when the aerosol signal is strong and toward very low values later, which appears to be the case in Figure 4. Kar et al. (2019) reported that to retrieve profiles of stratospheric aerosol extinction and backscatter coefficients reliably, they relied on substantial spatial and temporal averaging because of CALIPSO's poor signal-to-noise ratio in the stratosphere. Applied vertical averaging was 900 m, 5° latitude, and 20° longitude for a whole month. They averaged Level 1b backscattered profiles, not level 2 extinction profiles. Others used similar averaging to retrieve their stratospheric aerosol profiles (Thomason et al., 2007; Vernier et al., 2009; Vernier et al., 2011). The authors need to describe better the CALIPSO data and methods used to derive the sAOD. If indeed they used L2 data, they need to acknowledge its limitation when comparing it to OMPS and the model's sAOD.

Apologies – The description of the data and methods to derive the sAOD have been updated for clarity. You are correct in your assumption of which data product we use. Its limitation when comparing to OMPS and the model sAOD will be acknowledged in the relevant sections.

L119: "Campbell et al. (2012)". The reference is missing.

Corrected

L152: "80km", change to 40.5km.

Changed.

L164: "The retrieval issues described here have a significant impact on our study since it was estimated that the initial plume reached altitudes of between 11 and 20km (Vaughan et al., 2021; Osborne et al., 2021)." This sentence contradicts the previous one, which states that the loss of sensitivity below 17 km is only for the short wavelength and SH. Please explain why this is an issue in the NH where this eruption took place.

Apologies, that was not clear. There is loss of sensitivity below 17km everywhere, but this is more pronounced in the SH. Text changed for clarity.

L293: "The notable difference between the observations and the model is unexpected given that the magnitude of SO2 injected was based on observations (Muser et al., 2020; Kloss et al., 2021; De Leeuw et al., 2021)." Have you considered that the modeled $SO_2$ might compare better with other instruments or datasets, given that cited $SO_2$ volcanic mass was derived using different instruments?

Thank you, we have considered this and the text has been updated to reflect that.

Sections 4.2 and 4.3, Figures 4 and 5s: If the authors used L2 CALIPSO data (see my previous comment), they need to revise this section. OMPS LP and the model provide complete aerosol profiles, unlike CALIPSO L2, which only provides the elevated layers. I also suggest they compare their sAOD to CALIPSO's official L3 data for reference.

Thank you. This section has been revised to highlight the differences between the CALIOP data and the OMPS-LP data. Unfortunately the official L3 data cannot be utilised here due to a lack of data around this time frame. Making comparisons of CALIOP and the model sAOD is more consistent than comparing CALIOP and OMPS-LP as the model has the CALIOP limits imposed for consistency. This means that the model – in this case – also has incomplete aerosol profiles and is therefore creating the same bias as the observations. Whilst this is not a perfect comparison, we believe this is the best option without starting from scratch with a new dataset.

Section 4.6 and Figure 9 and L520: "After this, the observations show much of the aerosol plume below the average tropopause height, and by December, the magnitude of aerosol extinction coefficient is negligible."
Simplifying the volcanic aerosol distribution and tropopause altitudes in NH into one profile/one altitude per month can lead to the wrong conclusions. The volcanic plume is transported poleward to lower altitudes but still in the stratosphere, which is not apparent in the figure. The tropopause altitudes can vary significantly between 30-90N (~7 and 15km). Therefore, the conclusion that the plume is moving to the troposphere is inaccurate. The authors should modify the figure by removing the tropospheric part of each profile before calculating the monthly mean profile, thus ensuring that it only represents the stratospheric profile. Similarly, they can also calculate the tropospheric profile if they

believe that a significant part of the plume was transported to the troposphere, although I doubt it. They should also modify the text accordingly.

We agree with the arguments made by the reviewer. As a result, we have included a new figure which shows the monthly aerosol extinction coefficient as a function of latitude and altitude. This gets round the problem of showing a single line for the height of the tropopause (albeit with the variability also indicated). In this figure we do see some transfer of aerosol from the stratosphere to the troposphere, mostly at the midlatitudes. This section has been rewritten to include the new analysis.

L533: "This could be due to aerosol microphysical processes such as the rate of coagulation and/or condensation or how the model represents new particle formation." I don't understand why the authors are speculating on particle growth. They should be able to investigate the particle size in the model and provide a definitive explanation.

We agree that this statement was speculative. Section 4.6 has been rewritten to include new analysis.

L549: "....Due to more removal processes in the troposphere this results in a much faster decay rate and shorter lifetime." See my previous comment. Please revise this paragraph and any discussion regarding the average tropopause altitude.

We have updated the text.

Section 4.7 Radiative impact: This section is a significant result of this work; however, the way it is presented is lacking. The RI calculation is only shown for July when Raikoke's sAOD peaked in the later months. The authors need to expand their analysis and table 2 to provide the RI for other months. The analysis did not show when Sarychev's sAOD peaked. Was it in July or later?

Thank you. The analysis has now been updated to a longer record. Data for the Sarychev eruption is only available for three months (Haywood et al., 2010). However, the reviewer is correct that there are some interesting findings. Getting to the bottom of the differences between the two simulations is hampered by differences in the base-model, the aerosol scheme, vertical profiles, emissions of SO2 and ash and meteorology.

L580: "We also note that we cannot make exact global comparisons between our analysis and the Sarychev Peak eruption due to the impacts of the first Ulawun eruption (26th June) in UKESM1" I don't understand why this is not possible. The authors should be able to run the model simulation without Ulawun and derive the Raikoke-only RI. Please revise the text accordingly.

Apologies, the analysis has now been updated to compare a Raikoke-only simulation against Sarychev. We can therefore make global comparisons and this has been updated in the text.

L584: "The cooling effect is greatest over North America, similar to the distribution of sAOD. This is in contrast to the Sarychev Peak eruption, where the modelled radiative impact

indicated a stronger cooling over Russia in comparison to North America." Can you please comment on the cause of this difference?

I would expect this is to do with the different meteorology and nudged conditions for each model. We are now comparing the two months where the eruptions are at their peak and we do not see this difference so this has been removed from the text. This could also be due to how the aerosol were dispersed over time, the Sarychev eruption occurred much earlier in the month of June compared to Raikoke.

Section 5. "Discussion and conclusion": the section should be renamed "Summary and conclusion" as it mainly summarizes the analysis and results presented in the previous sections.

Corrected

L642:" lower in altitude resulting in a much faster decay rate due to transfer to the troposphere through tropospheric folds." See my previous comment regarding the average tropopause and revise the text accordingly.

The new figure 10 does show that SO2only simulation is consistently lower in altitude than the observations and SO2+ash, with significant portions in the troposphere.

L655: "Some of these differences may stem from the large dependence on meteorology that has been noted for low-altitude eruptions (Jones et al., 2016)." I don't understand this explanation. The main difference between the two eruptions is the presence of a large amount of ash in Raikoke, not the meteorology. The authors should investigate their model for the exact differences in the aerosol optical properties that result in such differences and provide a better explanation. They could also compare the $SO_2$only RI to the $SO_2$+ash and Sarychev, which is mostly $SO_2$.

When we look at the SO2only model, we see similar differences. We have a greater NH AOD in comparison to Sarychev, although less than SO2+ash which is to be expected following our previous analysis. The radiative forcing increases in the SO2only model which is due to the lack of ash present. One of the biggest differences between these eruptions is the model, UKESM1 is an updated version of HadGEM2 and this has been acknowledged in the text.

L663: "Future work might consider ..." The authors should include the addition of smoke aerosol to future work needed to improve their results.

Added

The paper is missing the code and data availability section.

A code and data availability section has been added.